# Further Studies of the Supersymmetric NJL-Type Model for a Real Superfield Composite

**Yifan Cheng [1]** , **Yan-Min Dai [2]** , **Gaber Faisel [3]** and **Otto C. W. Kong [2,4,]***

[1]  Center for General Education, National Tsing Hua University, Hsinchu 300044, Taiwan; yifancheng@mx.nthu.edu.tw

[2]  Department of Physics, National Central University, Chung-Li 32054, Taiwan; musselmusselmuss@yahoo.com.tw

[3]  Department of Physics, Faculty of Arts and Sciences, Süleyman Demirel University, Isparta 32260, Turkey; gaberfaisel@sdu.edu.tr

[4]  Center for High Energy and High Field Physics, National Central University, Chung-Li 32054, Taiwan

[*]  Correspondence: otto@phy.ncu.edu.tw

**Abstract:** This is a sequel to our earlier paper presenting a supersymmetric Nambu–Jona–Lasinio (NJL)-type model for a real superfield composite. The model in the simplest version has only a chiral superfield (multiplet), with a strong four-superfield interaction in the Kähler potential that induces a real two-superfield composite with vacuum condensate. The latter can have supersymmetry breaking parts, which we have shown to bear nontrivial solutions under a standard nonperturbative analysis for a Nambu–Jona–Lasinio-type model on a superfield setting. In this article, we generalize our earlier analysis by allowing a supersymmetric mass term for the chiral superfield, as well as possible $\theta^2$ components for the soft supersymmetry breaking part of the condensate. We present admissible nontrivial vacuum solutions and an analysis of the resulted low energy effective theory with components of the composite becoming dynamical. The determinant of the fermionic modes is shown to be zero, illustrating the presence of the expected Goldstino.

**Keywords:** NJL model; superfield composite; supersymmetry

**PACS:** 11.10.Stbound and unstable states; Bethe–Salpeter equations; 11.15.Pg expansions for large numbers of components (e.g., 1/Nc expansions); 11.30.Pb Supersymmetry

## 1. Introduction

With the discovery of the Higgs particle at the large hadron collider (LHC), the full success of the standard model (SM) has been crowned. Unfortunately, we still do not see any clear indication of experimental features beyond, so long as phenomenology at the TeV scale is concerned. Theorists are however mostly unsatisfied with the SM, particularly with its Higgs sector and explanation of the origin of the electroweak symmetry breaking. With a negative mass-square at the right scale put in by hand, the Higgs mechanism looks like only a phenomenological description of the 'true' theory behind it. Moreover, the other parts of the SM theory have their field content tightly constrained by the gauge symmetry and no parameters with mass dimensions admissible; everything in the Higgs sector looks completely arbitrary in comparison. Another way of looking at the issue would be that the only natural value of any input mass parameter should be like the model cutoff scale. We may need a model with a dynamical mechanism to generate the extra mass scale substantially below the cutoff.

Practically and experimentally accessible physics is really only about effective (field) theories. Taking the SM as an effective field theory, one would admit the higher dimensional operators with

couplings suppressed by powers of the model cutoff scale in the Lagrangian. Actually, a dimension six term of four-fermion interaction with strong coupling gives interesting nonperturbative dynamics that can break symmetries and generate masses [1]. That is the Nobel prize-winning classic Nambu–Jona-Lasinio (NJL) model [2,3], to which Higgs physics may correspond to the low energy effective theory, with the Higgs boson being identified as a two-quark composite. This beautiful idea of the top-mode SM [4–9] fails to accommodate the too small phenomenological top quark mass [1]. At this point, it looks like a holomorphic supersymmetric version that gives the (minimal) supersymmetric standard model (SSM) with both Higgs supermultiplets as two-superfield composite maintains phenomenological viability [10].

The SSM is still the most popular candidate theory beyond the SM. The theoretical beauty of supersymmetry is certainly part of its appeal. While the naive picture of supersymmetry at or below the TeV scale as a solution to the 'hierarchy problem' fails, the dynamical electroweak symmetry breaking option of a supersymmetric Nambu–Jona–Lasinio (SNJL) model may work with a higher supersymmetry breaking scale. The first SNJL model was introduced in the early 1980s [11,12], generalizing the four-fermion interaction to a four-superfield interaction of the same dimension, in the Kähler potential. A holomorphic version (HSNJL) as an alternative supersymmetrization [13] with a four-superfield interaction in the superpotential [10] has been introduced, with different theoretical and phenomenological merits [10,14,15]. However, this kind of supersymmetric NJL-type model discusses only the formation of composite chiral superfields, for modeling the Higgs sector. There are more interesting possibilities in the supersymmetrization of NJL-type models. Here, we focus on a simple one giving a real two-superfield composite from a dimension-six four-chiral-superfield interaction, first reported in our earlier paper [16]. Following and extending the framework of our earlier analyses [14–16], we present, in the article, further interesting nontrivial solutions for the soft mass parameters of the composite, under a more general setting, and the resulting physics. This kind of model may have the chance of dynamically generating the SNJL model with soft supersymmetry breaking masses for the further dynamical electroweak symmetry breaking, though the simple NJL framework is not able to have a firmly conclusive result on the dynamical supersymmetry breaking beyond its nature of a large-N approximation.

In Section 2, we present the model and the supergraph derivation of the superfield gap equation, elaborating carefully the extension of our framework of analysis [14,15] with model parameters and correlation functions taken as superspace parameters, like constant superfields, containing supersymmetric and supersymmetry breaking parts. The superfield gap equation contains components which include wavefunction renormalization factor and two different soft mass parameters. In Section 3, we discuss the effective theory picture with the composite and the matching effective potential analysis performed at the component field level, further strengthen the result and illustrate the physics involved. Section 4 is devoted to the analysis of the nontrivial solutions for the soft supersymmetry breaking parameters. In Section 5, we go further to look at some dynamical features of the composite superfield or its various components at low energy, focusing on the Goldstino mode. Some remarks and conclusions will be presented in the last section. Two appendices are given, the first on some details of the analytical expressions as background for the effective theory analysis and some results for two-point functions of the various components of the composite superfield relevant for their low energy dynamics, and the second on propagator expressions for a (chiral) superfield and components admitting the most general mass parameters. The latter expressions have not been explicitly presented in the literature.

## 2. The Model and the Superfield Gap Equation

The model has a dimension six four-superfield interaction similar but somewhat different from that of the SNJL model [11,12]. We start with the single chiral superfield (multiplet) Lagrangian

$$\mathcal{L} = \int d^4\theta \left[ \bar{\Phi}\Phi + \frac{m_o}{2}\Phi\Phi\delta^2(\bar{\theta}) + \frac{m_o^*}{2}\bar{\Phi}\bar{\Phi}\delta^2(\theta) - \frac{g_o^2}{2}(\bar{\Phi}\Phi)^2 \right] , \tag{1}$$

where $\Phi$ is the chiral superfield which is a scalar field on the chiral superspace, and $\theta$ is the Grassmann number which is one coordinate of superspace. We have suppressed any multiplet (color) indices, with basic notation being in line with that of Wess and Bagger [17]. Note the two supersymmetric mass terms in the superpotential which are absent in our previous study [16]. We present here a standard NJL gap equation analysis [11,14,15], to check for the dynamical generation of a $\bar{\Phi}\Phi$ composite and the related physics.

Before getting into our analysis, some comments on the symmetry issues are in order. Apart from supersymmetry itself, the model Lagrangian has, independent of the multiplet content of $\Phi_a$, a $U(1)_R$, symmetry under which $\Phi_a$ has a unit charge. With vanishing $m_o$, it has a full $U(N)$ symmetry under which the multiplet can be considered in the fundamental representation [16]. With the condition lifted, $\Phi_a$ may then be considered as an $SO(N)$, instead of $SU(N)$, multiplet. It is important to note that the usual $1/N$ approximation picture can still be valid. The $U(1)$ $\Phi$-number symmetry is only violated by the mass term, in the Lagrangian. In the naive case of a single superfield, the gap equation analysis here would correspond to the quenched planar approximation of QED by Bardeen et al. [18–20], which is commonly believed to give the correct qualitative result in the kind of dynamical symmetry breaking studies. Some more discussion of the issue in a somewhat different setting is available in Reference [15].

Let us go onto a superfield gap equation analysis following and extending our earlier formulated framework [14,15]. We consider a superfield two-point proper vertex $\Sigma_{\Phi\Phi^\dagger}(p; \theta^2, \bar{\theta}^2)$, which can be treated as a constant superfield with components explicitly dependent on $\theta^2$ and $\bar{\theta}^2$. In the full superfield picture, $\Sigma_{\Phi\Phi^\dagger}(p; \theta^2, \bar{\theta}^2)$ can be expanded as

$$\Sigma_{\Phi_R\Phi_R^\dagger}(p; \theta^2\bar{\theta}^2) = \Sigma_r(p) - \Sigma_{\tilde{\eta}}(p)\theta^2 - \bar{\Sigma}_{\tilde{\eta}^*}(p)\bar{\theta}^2 - \Sigma_{\tilde{m}}(p)\theta^2\bar{\theta}^2 . \tag{2}$$

It contains different components. The supersymmetric part $\Sigma_r$ gives only a kinetic term, hence contributes to wavefunction renormalization. The part $\Sigma_{\tilde{m}}$ in itself is like a proper self-energy contribution to the scalar but not the fermion component, hence soft supersymmetry breaking. Note that with $\Phi = A + \sqrt{2}\psi\theta + F\theta^2$, the four-superfield interaction after the $d^4\theta$ integration, has the part $-g_o^2 AA^\dagger(\Phi\Phi^\dagger)|_{\theta,\bar{\theta}=0}$. We have also performed the calculation fully in the component field framework for cases of Reference [16], but prefer to illustrate the superfield calculation here in accordance with the formulation under the perspective discussed in Reference [14]. The soft supersymmetry breaking mass $\tilde{m}^2$ as a superfield term is just the $\theta^2\bar{\theta}^2$ component of the kinetic term, to which $\Sigma_{\Phi\Phi^\dagger}(p; \theta^2, \bar{\theta}^2)$ is the quantum correction to the latter. The part $\Sigma_{\tilde{\eta}}$, which is not considered in our earlier study [16], is somewhat less obvious. It is a proper vertex of $AF^*$, to be matched to another mass parameter $\tilde{\eta}$; the $\tilde{\eta}AF^*$ term gives another kind of soft supersymmetry breaking mass not usually discussed in the literature.

To keep notation simple, we will present our analysis here onwards with the index suppressed, as if we are working on a single superfield. What we have in mind is really a $N$-multiplet of the $SO(N)$ or $SU(N)$. Retrieving result for a nontrivial $N$ is straightforward. The one-loop contribution such as the one in $\Sigma_{\Phi\Phi^\dagger}(p; \theta^2, \bar{\theta}^2)$ or $\Sigma_{\tilde{m}}(p)$ will have to be multiplied by the factor $N$.

It is then easy to appreciate that a consistent superfield treatment of the standard NJL analysis should consider modifying the superfield propagator to incorporate plausible nonperturbative parameter of the generic form given by

$$\mathcal{Y} = y - \tilde{\eta}_o\theta^2 - \tilde{\eta}_o^*\bar{\theta}^2 - \tilde{m}_o^2\theta^2\bar{\theta}^2 \tag{3}$$

where containing supersymmetric as well as supersymmetry breaking parts. We write here $\tilde{m}_o^2$ and $\tilde{\eta}_o$ instead of $\tilde{m}^2$ and $\tilde{\eta}$, as the parameters are not physical ones yet. The component $y$ contributes a (supersymmetric) wavefunction renormalization factor which renormalizes all mass parameters accordingly, as shown below explicitly. Particularly, in the former work, [16] $\tilde{\eta}_o = 0$ was assumed for simplicity. However, we do not make such an assumption here to give a more complete picture of the analysis. Notice that a generation of nontrivial $y$ breaks no symmetry while a generation

of $\tilde{m}^2$ breaks only supersymmetry. Non-vanishing $\tilde{\eta}$ however breaks the $U(1)_R$ symmetry together with supersymmetry.

To proceed with the derivation of the superfield gap equation, we add and subtract the term $\mathcal{Y}\bar{\Phi}\Phi$, as the first step of the self-consistent Hartree approximation, and split the Lagrangian as $\mathcal{L} = \mathcal{L}_o + \mathcal{L}_{int}$ where

$$\mathcal{L}_o \;=\; \int d^4\theta \left[ \bar{\Phi}\Phi(1+\mathcal{Y}) + \frac{m_o}{2}\Phi^2\delta^2(\bar{\theta}) + \frac{m_o^*}{2}\bar{\Phi}^2\delta^2(\theta) \right] \tag{4}$$

and

$$\mathcal{L}_{int} = \int d^4\theta \left[ -\mathcal{Y}\bar{\Phi}\Phi - \frac{g_o^2}{2}\bar{\Phi}\Phi\bar{\Phi}\Phi \right] . \tag{5}$$

To restore the canonical kinetic term in the presence of a plausibly nonzero $y$, we introduce the renormalized superfield $\Phi_R \equiv \sqrt{Z}\Phi = \sqrt{1+y}\Phi$ which gives

$$\mathcal{L}_o \;=\; \int d^4\theta \left[ \bar{\Phi}_R\Phi_R(1 - \tilde{\eta}\theta^2 - \tilde{\eta}^*\bar{\theta}^2 - \tilde{m}^2\theta^2\bar{\theta}^2) + \frac{m}{2}\Phi_R^2\delta^2(\bar{\theta}) + \frac{m^*}{2}\bar{\Phi}_R^2\delta^2(\theta) \right] . \tag{6}$$

The mass parameters are of course renormalized ones, to be divided by the wavefunction renormalization parameter $Z$; explicitly $m = \frac{m_o}{1+y}$, for example. The quantum effective action is

$$\begin{aligned}
\Gamma \;=\;\; & \bar{\Phi}_R\Phi_R(1 - \tilde{\eta}\theta^2 - \tilde{\eta}^*\bar{\theta}^2 - \tilde{m}^2\theta^2\bar{\theta}^2) + \frac{m}{2}\Phi_R^2\delta^2(\bar{\theta}) + \frac{m^*}{2}\bar{\Phi}_R^2\delta^2(\theta) \\
& -\mathcal{Y}_R\bar{\Phi}_R\Phi_R - \frac{g^2}{2}\bar{\Phi}_R\Phi_R\bar{\Phi}_R\Phi_R + \Sigma_{\Phi_R\Phi_R^\dagger}\bar{\Phi}_R\Phi_R + \cdots ,
\end{aligned} \tag{7}$$

where $g^2 = \frac{g_o^2}{(1+y)^2}$ is the renormalized four-superfield coupling and $\mathcal{Y}_R$ is similarly given by

$$\mathcal{Y}_R = \frac{\mathcal{Y}}{Z} = \frac{y}{1+y} - \tilde{\eta}\theta^2 - \tilde{\eta}^*\bar{\theta}^2 - \tilde{m}^2\theta^2\bar{\theta}^2 . \tag{8}$$

The superfield gap equation under the NJL framework is then given by

$$-\mathcal{Y}_R + \Sigma_{\Phi_R\Phi_R^\dagger}^{(loop)}(p; \theta^2\bar{\theta}^2)\Big|_{\text{on-shell}} = 0 . \tag{9}$$

In component form, we have

$$\begin{aligned}
\frac{y}{1+y} &= \left. \Sigma_r^{(loop)}(p) \right|_{\text{on-shell}} , \\
\tilde{\eta} &= \left. \Sigma_{\tilde{\eta}}^{(loop)}(p) \right|_{\text{on-shell}} , \\
\tilde{m}^2 &= \left. \Sigma_{\tilde{m}^2}^{(loop)}(p) \right|_{\text{on-shell}} ,
\end{aligned} \tag{10}$$

where, in accordance with the standard NJL analysis, one uses the one-loop contribution to $\Sigma_{\Phi_R\Phi_R^\dagger}(p; \theta^2\bar{\theta}^2)$ from the four-superfield interaction. The diagrammatic illustration of the renormalized superfield gap equation is given in Figure 1. Note that results reported in Reference [16] correspond to assuming $\tilde{\eta}$ remains zero from the beginning, which will be shown to be a consistent solution. Our interest here is on solutions with nontrivial $\tilde{\eta}$.

We perform a supergraph calculation for $\Sigma_{\Phi_R \Phi_R^\dagger}(p; \theta, \bar{\theta})$ directly. The relevant superfield propagator is given by

$$
\begin{aligned}
\langle T(\Phi(1)_R \Phi_R^\dagger(2)) \rangle &= \frac{-i}{p^2 + |m|^2} \delta_{12}^4 - i\frac{\tilde{\eta}(Q - 2|m|^2)}{Q^2 - 4|m|^2|\tilde{\eta}|^2} \theta_1^2 \delta_{12}^4 - i\frac{\tilde{\eta}^*(Q - 2|m|^2)}{Q^2 - 4|m|^2|\tilde{\eta}|^2} \bar{\theta}_1^2 \delta_{12}^4 \\
&\quad + i\frac{(\tilde{m}^2 + |\tilde{\eta}|^2)Q - 4|m|^2|\tilde{\eta}|^2}{(p^2 + |m|^2)(Q^2 - 4|m|^2|\tilde{\eta}|^2)} \left[\frac{D_1^2 \theta_1^2 \bar{\theta}_1^2 \overline{D}_1^2}{16}\right] \delta_{12}^4 \\
&\quad + i\frac{(-p^2|\tilde{\eta}|^2 + \tilde{m}^2|m|^2)Q + 4p^2|m|^2|\tilde{\eta}|^2}{(p^2 + |m|^2)(Q^2 - 4|m|^2|\tilde{\eta}|^2)} \theta_1^2 \bar{\theta}_1^2 \delta_{12}^4 \,,
\end{aligned}
\tag{11}
$$

where $Q = p^2 + |m|^2 + |\tilde{\eta}|^2 + \tilde{m}^2$ and $\delta_{12}^4 = \delta^4(\theta_1 - \theta_2)$. The necessary evaluation of $\Sigma_{\Phi_R \Phi_R^\dagger}^{(loop)}(p; \theta^2 \bar{\theta}^2)\big|_{\text{on-shell}}$ is much similar to previous cases [14]. The result is given by

$$
\begin{aligned}
\Sigma_{\Phi_R \Phi_R^\dagger}^{(loop)}(p; \theta^2 \bar{\theta}^2)\Big|_{\text{on-shell}} &= -g^2 \int^E \left[ \frac{1}{k^2 + |m|^2} + \frac{\tilde{\eta}(Q_k - 2|m|^2)}{Q_k^2 - 4|m|^2|\tilde{\eta}|^2} \theta^2 + \frac{\tilde{\eta}^*(Q_k - 2|m|^2)}{Q_k^2 - 4|m|^2|\tilde{\eta}|^2} \bar{\theta}^2 \right. \\
&\quad - \frac{(\tilde{m}^2 + |\tilde{\eta}|^2)Q_k - 4|m|^2|\tilde{\eta}|^2}{(k^2 + |m|^2)(Q_k^2 - 4|m|^2|\tilde{\eta}|^2)} \left(1 - k^2 \theta^2 \bar{\theta}^2 + 4k_a \sigma_{\alpha\dot{\alpha}}^a \theta^\alpha \bar{\theta}^{\dot{\alpha}}\right) \\
&\quad \left. - \frac{(-k^2|\tilde{\eta}|^2 + \tilde{m}^2|m|^2)Q_k + 4k^2|m|^2|\tilde{\eta}|^2}{(k^2 + |m|^2)(Q_k^2 - 4|m|^2|\tilde{\eta}|^2)} \theta^2 \bar{\theta}^2 \right] \,,
\end{aligned}
\tag{12}
$$

where the $\int^E$ denotes integration over Euclidean four-momentum $k$ with the measure $\frac{d^4k}{(2\pi)^4}$ and $Q_k = k^2 + |m|^2 + |\tilde{\eta}|^2 + \tilde{m}^2$. Each of the five terms in the above expression comes exactly from the corresponding term in the superfield propagator. The $4k_a \sigma_{\alpha\dot{\alpha}}^a \theta^\alpha \bar{\theta}^{\dot{\alpha}}$ term vanishes upon integration. The others can be pull together to give the component gap equations as

$$
\begin{aligned}
\frac{y}{1+y} &= \Sigma_r^{(loop)}(p)\Big|_{\text{on-shell}} = -g^2 \int^E \frac{(k^2 + |m|^2 + \tilde{m}^2 + |\tilde{\eta}|^2)}{(k^2 + |m|^2 + \tilde{m}^2 + |\tilde{\eta}|^2)^2 - 4|m|^2|\tilde{\eta}|^2} \,, \\
\tilde{\eta} &= \Sigma_{\tilde{\eta}}^{(loop)}(p)\Big|_{\text{on-shell}} = g^2 \tilde{\eta} \int^E \frac{(k^2 - |m|^2 + \tilde{m}^2 + |\tilde{\eta}|^2)}{(k^2 + |m|^2 + \tilde{m}^2 + |\tilde{\eta}|^2)^2 - 4|m|^2|\tilde{\eta}|^2} \,, \\
\tilde{m}^2 &= \Sigma_{\tilde{m}^2}^{(loop)}(p)\Big|_{\text{on-shell}} = g^2 \int^E \frac{1}{(k^2 + |m|^2)} \frac{1}{(k^2 + |m|^2 + \tilde{m}^2 + |\tilde{\eta}|^2)^2 - 4|m|^2|\tilde{\eta}|^2} \\
&\quad \cdot \left\{ \left[\tilde{m}^2(k^2 - |m|^2) + 2k^2|\tilde{\eta}|^2\right](k^2 + |m|^2 + \tilde{m}^2 + |\tilde{\eta}|^2) - 8k^2|m|^2|\tilde{\eta}|^2 \right\} \,.
\end{aligned}
\tag{13}
$$

Non-trivial solutions of the three coupled equations with nonvanishing $\tilde{\eta}$ and/or $\tilde{m}^2$ give supersymmetry breaking solutions, while nontrivial $y$ value gives wavefunction renormalization to $\Phi$, which does not change the qualitative answer to whether supersymmetry breaking solutions with the soft mass generation exist. Our analysis will explicitly demonstrate that. However, we postpone the analysis of the nontrivial solution until after the discussion of the effective theory picture in the next section.

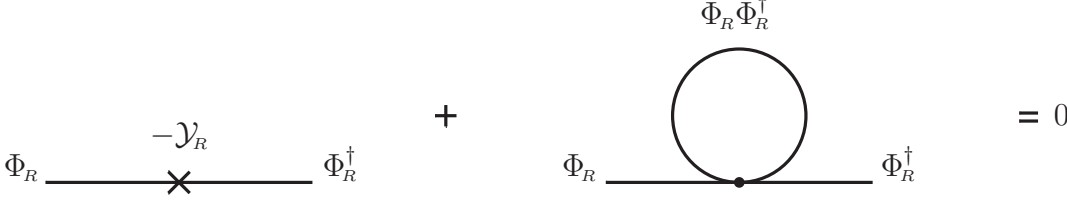

**Figure 1.** The renormalized superfield gap equation, with $\mathcal{Y}_R = \frac{y}{1+y} - \tilde{\eta}\theta^2 - \tilde{\eta}^*\bar{\theta}^2 - \tilde{m}^2\theta^2\bar{\theta}^2$.

### 3. The Effective Theory Picture

Following the general effective theory picture of the NJL-type models, we modify the model Lagrangian by adding to it

$$\mathcal{L}_s = \int d^4\theta \frac{1}{2}(\mu U + g_o \bar{\Phi}\Phi)^2 \,, \tag{14}$$

where $U$ is an 'auxiliary' real superfield and the mass parameter $\mu$ is taken as real and positive (for $g_o^2 > 0$). The equation of motion for $U$, from the full Lagrangian $\mathcal{L} + \mathcal{L}_s$ gives

$$U = -\frac{g_o}{\mu}\bar{\Phi}\Phi \,, \tag{15}$$

where showing it as a superfield composite of $\bar{\Phi}$ and $\Phi$. The condition says the model with $\mathcal{L} + \mathcal{L}_s$ is equivalent to that of $\mathcal{L}$ alone. Expanding the term in $\mathcal{L}_s$, we have a cancellation of the dimension six interaction in the full Lagrangian, giving it as

$$\mathcal{L}_{eff} \equiv \mathcal{L} + \mathcal{L}_s = \int d^4\theta \left[ \bar{\Phi}\Phi + \frac{\mu^2}{2}U^2 + \mu g_o U\bar{\Phi}\Phi + \frac{m_o}{2}\Phi^2\delta(\bar{\theta}) + \frac{m_o^*}{2}\bar{\Phi}^2\delta(\theta) \right] \,. \tag{16}$$

Obviously, if $U|_D$ develops a vacuum expectation value (VEV), supersymmetry is broken spontaneously and the superfield $\Phi$ gains a soft supersymmetry breaking mass of $\tilde{m}_o^2 = -\mu g_o \langle U|_D \rangle$. The above looks very much like the standard features of NJL-type model. Notice that while $U$ does contain a vector component, its couplings differ from that of the usually studied 'vector superfield' which is a gauge field supermultiplet. That is in addition to having $\mu$ as like a supersymmetric mass for $U$, which can be compatible only with a broken gauge symmetry. As such, the model with superfield $U$ is not usually discussed. The superfield can be seen as two parts, as illustrated by the following component expansion,

$$\begin{aligned}
U(x,\theta,\bar{\theta}) &= \frac{C(x)}{\mu} + \sqrt{2}\theta\frac{\chi(x)}{\mu} + \sqrt{2}\bar{\theta}\frac{\bar{\chi}(x)}{\mu} + \theta\theta\frac{N(x)}{\mu} + \bar{\theta}\bar{\theta}\frac{N^*(x)}{\mu} \\
&\quad + \sqrt{2}\theta\sigma^\mu\bar{\theta}v_\mu(x) + \sqrt{2}\theta\theta\bar{\theta}\bar{\lambda}(x) + \sqrt{2}\bar{\theta}\bar{\theta}\theta\lambda(x) + \theta\theta\bar{\theta}\bar{\theta}D(x) \,,
\end{aligned} \tag{17}$$

where the components $C$, $\chi$, and $N$ are the first parts which have content like a chiral superfield with, however, $C$ being real. The $\mu$ factor is put to set the mass dimensions right. The rest is like the content of a superfield for the usual gauge field supermultiplet, with $D$ and $v_\mu$ real. The effective Lagrangian in component form is given by

$$\begin{aligned}
\mathcal{L}_{eff} &= (1 + g_o C)\left[A^*\Box A + i(\partial_\mu\bar{\psi})\bar{\sigma}^\mu\psi + F^*F\right] + \frac{m_o}{2}(2AF - \psi\psi) + \frac{m_o^*}{2}(2A^*F^* - \bar{\psi}\bar{\psi}) \\
&\quad + \mu CD - \mu\chi\lambda - \mu\bar{\chi}\bar{\lambda} + NN^* - \frac{\mu^2}{2}v^\nu v_\nu - \mu g_o\psi\lambda A^* - \mu g_o\bar{\psi}\bar{\lambda}A + \mu g_o DA^*A \\
&\quad - i\frac{g_o}{2}\bar{\psi}\bar{\sigma}^\mu\chi\partial_\mu A + i\frac{g_o}{2}(\partial_\mu\bar{\psi})\bar{\sigma}^\mu\chi A - g_o\chi\psi F^* + g_o NAF^* \\
&\quad + i\frac{g_o}{2}\bar{\chi}\bar{\sigma}^\mu\psi\partial_\mu A^* - i\frac{g_o}{2}A^*\bar{\chi}\bar{\sigma}^\mu\partial_\mu\psi - g_o\bar{\chi}\bar{\psi}F + g_o N^*A^*F \\
&\quad - \frac{\mu g_o}{\sqrt{2}}\eta^{\mu\nu}v_\mu iA^*\partial_\nu A + \frac{\mu g_o}{\sqrt{2}}\eta^{\mu\nu}v_\mu i(\partial_\nu A^*)A - \frac{\mu g_o}{\sqrt{2}}\eta^{\mu\nu}v_\mu\bar{\psi}\bar{\sigma}_\nu\psi \,.
\end{aligned} \tag{18}$$

Notice that, like $F$, $N$ and $D$ have mass dimension two.

Under the $U(1)_R$ symmetry, $A$ and $F$ have charge $+1$ and $-1$. The superfield $U$ is uncharged. However, components $N$, $\chi$ and $\lambda$ carry nontrivial $U(1)_R$ charges $-2$, $-1$ and $+1$, respectively. For the $m_o = 0$ case, there is an extra $U(1)$ $\Phi$-number symmetry with common charge for all components. All components of $U$ are not charged under the latter.

In accordance with the 'quark-loop' approximation in the (standard) NJL gap equation analysis and our particular supergraph calculation scheme above, we consider plausible non-trivial vacuum solution with non-zero vacuum expectation values (VEVs) for the composite scalars $C$, $D$ and $N$. While $N$ is complex, we can safely take $n \equiv \langle N \rangle$ to be real here. At least we can exploit the $U(1)_R$ symmetry to absorb any phase at the expense of having a complex $m_o$, the phase of which does not show up in the calculation. First note that scalar $C$ couples to kinetic terms of components of $\Phi$; $c \equiv \langle C \rangle$ hence contributes to a supersymmetric wavefunction renormalization of the latter. It is the supersymmetric part of $\Sigma_{\Phi\Phi^\dagger}^{(loop)}(p; \theta^2, \bar{\theta}^2)$, an unavoidable part of the one-loop supergraph in our gap equation calculation in the previous section. Again, we should go to the renormalized superfield $\Phi_R = \sqrt{(1 + g_o c)} \Phi$ in the following calculations, with renormalized mass $m$ and coupling $g$. With $n \equiv \langle N \rangle$ and $d \equiv \langle D \rangle$, we have $-gn$ and $-\mu g d$, corresponding to the supersymmetry breaking masses $\tilde{\eta}$ and $\tilde{m}^2$ of $\Phi_R$. In the former case, it gives a $A_R F_R^*$ component term. Note that $\langle N \rangle$ is the only VEV that breaks the $U(1)_R$ symmetry, as $C$ and $D$ carry no charges, though both $\langle N \rangle$ and $\langle D \rangle$ break supersymmetry.

With propagators for the components of the renormalized 'quark' superfield $\Phi_R$, as given in the Appendix A, one can easily obtain the minimum condition for the effective potential following the Weinberg tadpole method [21,22]. Firstly, for $C$-tadpoles, we have a $\Phi_R$ loop or in component form one from each of $A_R$, $\psi_R$, and $F_R$. Hence, we have up to one loop level

$$\Gamma_C^{(1)} = \Gamma_C^{(1)\text{tree}} + \Gamma_{C_A}^{(1)} + \Gamma_{C_\psi}^{(1)} + \Gamma_{C_F}^{(1)} = \mu d - g I_C , \tag{19}$$

where

$$
\begin{aligned}
I_C &= I_{C_A} - 2I_{C_\psi} + I_{C_F} ; \\
I_{C_A} &= \int^E \frac{k^2(k^2 + |m|^2 + g^2|n|^2 - \mu g d)}{(k^2 + |m|^2 + g^2|n|^2 - \mu g d)^2 - 4g^2|n|^2|m|^2} , \\
I_{C_\psi} &= \int^E \frac{k^2}{k^2 + |m|^2} , \\
I_{C_F} &= \int^E \frac{(k^2 - \mu g d)(k^2 + |m|^2 + g^2|n|^2 - \mu g d)}{(k^2 + |m|^2 + g^2|n|^2 - \mu g d)^2 - 4g^2|n|^2|m|^2} .
\end{aligned}
\tag{20}
$$

Next, the $N^*$-tadpole is given by

$$\Gamma_{N^*}^{(1)} = n - g I_N , \tag{21}$$

where

$$I_N = \int^E \frac{gn(k^2 - |m|^2 + g^2|n|^2 - \mu g d)}{(k^2 + |m|^2 + g^2|n|^2 - \mu g d)^2 - 4g^2|n|^2|m|^2} . \tag{22}$$

The $D$-tadpole is given by

$$\Gamma_D^{(1)} = \mu c + \mu g I_D \tag{23}$$

where

$$I_D = \int^E \frac{k^2 + |m|^2 + g^2|n|^2 - \mu g d}{(k^2 + |m|^2 + g^2|n|^2 - \mu g d)^2 - 4g^2|n|^2|m|^2} . \tag{24}$$

The tadpole diagrams are illustrated in Figure 2. We look for vacuum solution with $-\Gamma_a^{(1)} \equiv \partial V(c, n, d)_{\text{1-loop}}/\partial a = 0$ for $a = c, n, d$. Firstly, note that the vanishing of $N^*$-tadpole is equivalent to

$$n(1 - g^2 I_{N'}) = 0 , \tag{25}$$

with $I_{N'}$ given by $I_N = gnI_{N'}$; vanishing $D$-tadpole gives

$$c = -gI_D ; \tag{26}$$

the vanishing $C$-tadpole condition is

$$\mu d = gI_C . \tag{27}$$

To get the physics picture clear, one can identify the soft masses generated for the superfield $\Phi$ by $\tilde{\eta} = -gn$ and $\tilde{m}^2 = -\mu gd$. We will explore nontrivial solutions for the soft masses below.

It is interesting to see that the effective potential analysis for (the components of) the composite superfield $U$ can be shown directly to be equivalent to the superfield gap equation, which we illustrated explicitly in Reference [16] and duplicated here. In terms of the superfield, the potential minimum condition is given by

$$\mu^2 \langle U \rangle + U_{tadpole} = 0 \quad \implies \quad \mu g \langle U \rangle = -g^2 I_{\Phi_R \Phi_R^\dagger}^{(loop)} \tag{28}$$

where $I_{\Phi_R \Phi_R^\dagger}^{(loop)}$ is the momentum integral of the $\Phi_R \Phi_R^\dagger$ propagator loop (*cf.* the first diagram in Figure 2). Note that from the original Lagrangian with two-superfield composite assumed, we can obtain $-g^2 \langle (\Phi_R \Phi_R^\dagger) \rangle = \mathcal{Y}_R$, which is equivalent to $\mu g \langle U \rangle = \mathcal{Y}_R = \Sigma_{\Phi_R \Phi_R^\dagger}^{(loop)}(p; \theta^2 \bar{\theta}^2)\Big|_{\text{on-shell}} = -g^2 I_{\Phi_R \Phi_R^\dagger}^{(loop)}$. The same loop integral is of course involved in both the gap equation picture and the effective potential analysis. The results here are in direct matching with the corresponding discussion for the NJL case presented in Reference [12], though for a superfield theory instead. The component field effective potential analysis above really serves as a double-check of the superfield gap equation analysis of the previous section. In terms of component fields, we need the soft mass identifications above as well as $y = g_o c$, or $\frac{y}{1+y} = gc$.

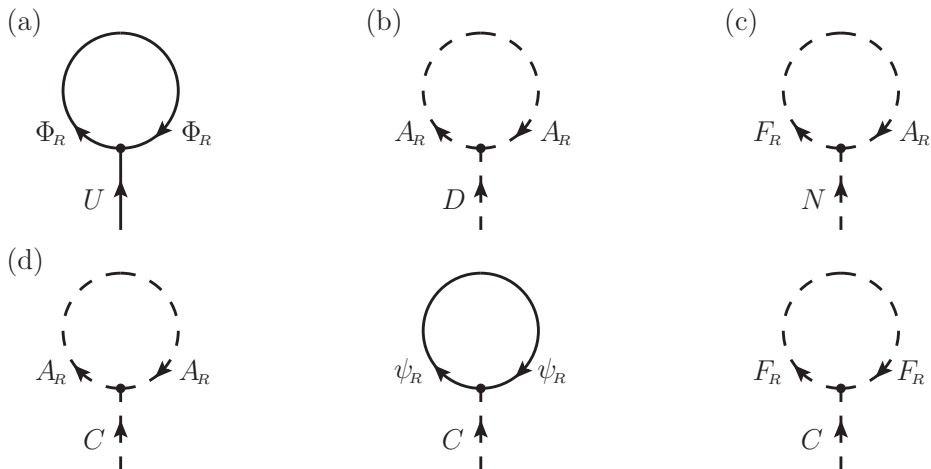

**Figure 2.** The tadpole diagrams: (**a**) the superfield diagram; (**b**) $D$-tadpole; (**c**) $N$-tadpole; (**d**) $C$-tadpoles.

## 4. Nontrivial Solutions

Let us pull together the gap equations in terms of $y = (1 - gc)^{-1}$, $\tilde{\eta}(= -gn)$ and $\tilde{m}^2(= -\mu gd)$. We have

$$\tilde{\eta}(1 - g^2 I'_N) = 0 \qquad \text{and} \qquad \tilde{m}^2 = -g^2 I_C \,,$$

as a set of coupled equations to be solved simultaneously. The integrals are complicated expressions involving the two soft mass parameters. The third equation of

$$y = (1 + g^2 I_D)^{-1}$$

where independently gives the $y$ value for any solution of $\tilde{\eta}$ and $\tilde{m}$. The trivial zero soft masses solution is consistent, as $I_C$ vanishes in the supersymmetric limit. Note that the $y$ parameter does not correspond to any physical quantity and hence may be considered of little interest. The point of interest is if the nontrivial solutions of supersymmetry breaking masses $\tilde{\eta}$ and $\tilde{m}$ exist.

The first equation above gives $g^2 I'_N = 1$ for non-trivial $\tilde{\eta}$, for the case of which we have

$$I'_N = \frac{1}{2}\left[\left(1 - \frac{|m|}{|\tilde{\eta}|}\right) I_F(m^2_{A_-}) + \left(1 + \frac{|m|}{|\tilde{\eta}|}\right) I_F(m^2_{A_+})\right], \tag{29}$$

where $I_F(S) \left[\equiv \int^E \frac{1}{k^2 + S}\right]$ has been used to denote integral of the Feynman propagator for field of mass square $S$ and we have the scalar mass eigenvalues

$$m^2_{A_\mp} = \tilde{m}^2 + (|m| \mp |\tilde{\eta}|)^2 \,. \tag{30}$$

As seen here, the presence of non-zero $m\tilde{\eta}$ product splits the masses of the scalar and pseudoscalar part of $A$ and produces mass mixing between them, giving the mass eigenvalues. The interesting point is that one only needs non-zero $m\tilde{\eta}$ to have it; even a real value would do.

Similarly, we have

$$I_C = -\left(m^2_{A_-} - \frac{\tilde{m}^2}{2}\right) I_F(m^2_{A_-}) - \left(m^2_{A_+} - \frac{\tilde{m}^2}{2}\right) I_F(m^2_{A_+}) + 2|m|^2 I_F(|m|^2) \,. \tag{31}$$

If we take $m = 0$, we would have

$$I'_N \longrightarrow I_F(|\tilde{\eta}|^2 + \tilde{m}^2)$$

and

$$I_C \longrightarrow -\tilde{m}^2 I_F(|\tilde{\eta}|^2 + \tilde{m}^2) - 2|\tilde{\eta}|^2 I_F(|\tilde{\eta}|^2 + \tilde{m}^2) \,.$$

The second soft mass gap equation becomes

$$g^2 I_F(|\tilde{\eta}|^2 + \tilde{m}^2)\left(1 + 2\frac{|\tilde{\eta}|^2}{\tilde{m}^2}\right) = 1 \tag{32}$$

which is not compatible with the first one ($g^2 I'_N = 1$) unless $\tilde{\eta} = 0$. It remains to be seen if there exist $\tilde{\eta} \neq 0$ solutions for some non-zero values of $m$. After some algebra, one can rewrite the solution equations in the form

$$g^2 I_F(m^2_{A_\mp}) = \frac{|m|\tilde{m}^2 \mp 2|\tilde{\eta}|(|m| \pm |\tilde{\eta}|)^2}{|m|(2|m|^2 - 2|\tilde{\eta}|^2 + \tilde{m}^2)} + \frac{2|m|(|m| \pm |\tilde{\eta}|)}{2|m|^2 - 2|\tilde{\eta}|^2 + \tilde{m}^2} g^2 I_F(|m|^2) \,. \tag{33}$$

The two equations have the same form only, with the $|\tilde{\eta}|$ variable coming in different signs. Moreover, both reduce to the same equation for the $I_F(m_A)$ at the $|\tilde{\eta}| = 0$ limit, which is the gap equation for the limiting case [16]. Evaluating the integrals with model cutoff $\Lambda$, with all variables and

parameters casted in terms of dimensionless counterparts normalized to $\Lambda$ given by $G = \frac{g^2\Lambda^2}{16\pi^2}$, $s = \frac{\tilde{m}^2}{\Lambda^2}$, and $t = \frac{|m|^2}{\Lambda^2}$, the two equations are equivalent to

$$\frac{1}{G(s,t,z)} = \frac{s + 2tz(1-z)}{s + 2tz(1-z)^2} + \frac{2t(1-z)}{s + 2tz(1-z)^2} t \ln\left[1 + \frac{1}{t}\right]$$
$$- \frac{s + 2t(1-z^2)}{s + 2tz(1-z)^2}[s + t(1+z)^2] \ln\left[1 + \frac{1}{s + t(1+z)^2}\right], \tag{34}$$

for $z = \mp\frac{|\tilde{\eta}|}{|m|}$, respectively. We need simultaneous solutions for $s$ and $z$ for reasonable values of model parameters $G$ and $t$. The two equations for positive and negative (but equal) values of $z$ of course collapse to one at $z = 0$, which are the vanishing $|\tilde{\eta}|$ solutions. We present them in Figure 3 for different values of supersymmetric mass $m$, among which the case of $m = 0$ has been illustrated in Reference [16].

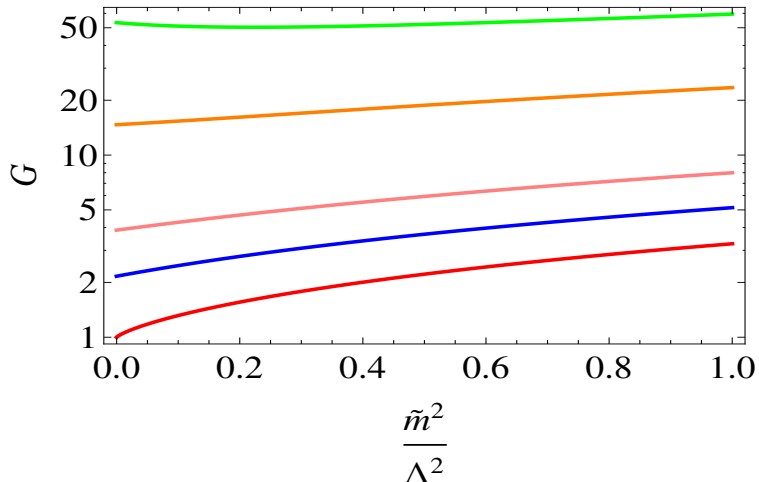

**Figure 3.** Numerical plot of nontrivial solutions to the soft mass gap equation with $|\tilde{\eta}| = 0$. Coupling parameter $G = \frac{Ng^2\Lambda^2}{16\pi^2}$ is plotted against the normalized soft mass parameter $s \left(= \frac{\tilde{m}^2}{\Lambda^2}\right)$ for $t \left(= \frac{|m|^2}{\Lambda^2}\right)$ values of 0 (red), 0.1 (blue), 0.2 (pink), 0.4 (orange), 0.5 (green), from the lowest to the highest curves, respectively. The red curve ($t = 0$) has been reported and analyzed in Reference [16]. Here, $N$ is the 'color' factor for the case of the basic chiral superfield $\Phi$ being an $SO(N)$ or $SU(N)$ multiplet not shown explicitly in the calculation, and $\Lambda$ is the model cutoff scale. Notice that the critical coupling increases from $G = 1$ for nonzero values of the input supersymmetric mass $m$.

Actually, in the $\tilde{\eta} = 0$ ($z = 0$) case, all the above integrals simplify analytically. In particular, we have

$$I_D \longrightarrow I_F(|m|^2 + \tilde{m}^2)$$

and

$$I_C \longrightarrow -(\tilde{m}^2 + 2|m|^2)I_F(|m|^2 + \tilde{m}^2) + 2|m|^2 I_F(|m|^2)$$

where ($I'_N$ is irrelevant). The masses in the Feynman propagators correspond to the scalar and fermion masses. It is interesting to note that for $m = 0$ ($t = 0$), we have the simple result $g^2 I_F(\tilde{m}^2) = 1$, which is the same as the basic NJL model one except with the soft mass $\tilde{m}^2$ replacing the (Dirac) fermionic mass (see for example Reference [12]) if we take $\frac{g^2}{2}$ as the four-fermion coupling in the model. Moreover, in this case, we have the gap equation for the renormalization factor, which is equivalent to the vanishing $D$-tadpole condition $c = -g I_D$, giving $c = -g I_F(\tilde{m}^2)$ hence $g_o c = -0.5$. The wavefunction renormalization factor is $Z = 1 + g_o c = 0.5$, of order one but tangible. This is a clear indication of the nonperturbative nature of the results and that there is nothing improper in the

analysis. For more details, we see that solutions for nontrivial $\tilde{m}^2$ for the case are given by the reduced form of Equation (34) as $\frac{1}{G} = 1 - s \ln\left[1 + \frac{1}{s}\right]$ obviously giving solution for $0 < s < 1$ for the strong enough coupling $G > 1$. It can be seen from the numerical plot that the value of the $\tilde{m}^2$ solution rises fast with increasing $G$. However, nonzero $t$ has a strong limiting effect. It increases the critical coupling needed for a nontrivial solution to $s$ very substantially. In fact, taking the limit $s \to 0$, the equation becomes $\frac{1}{G_c} = 1 + \frac{2t}{t+1} - 3t \ln\left(1 + \frac{1}{t}\right)$, which gives the critical coupling $G_c$ as a function of $t$. It can be seen then as $t$ increases from zero, $\frac{1}{G_c}$ decreases and reaches vanishing value ( i.e., $G_c \to \infty$) at a critical $t$ value of about 0.55, beyond which no coupling $G$ will be strong enough to break the supersymmetry and generate the soft mass.

Looking for solutions with nontrivial $\tilde{\eta}$ is more tricky and requires a very careful analysis scanning the numerical results. Again, we check plots of the effective coupling $G$ as given in Equation (34) versus $s$ simultaneously for positive and negative values of $z$ of fixed magnitude, at a fixed input $t$ value. Numerically, within the window of interest, the intersection of the two curves (dubbed $G_+$ and $G_-$, respectively) gives a solution. Only one then has to numerically scan the plots of the $G_+$ and $G_-$ curves to see all the solutions. The window of interest is restricted to positive $G$ value and $0 < s \leq 1$ plus the extra constraint of both of the mass eigenvalues of the scalar states in $\Phi$ to be within the cutoff $\Lambda$ [*cf.* Equation (30)]. This is the generalization of $s \leq 1$ to the nontrivial $\tilde{\eta}$, $|z| \neq 0$ case. The constraint is given by

$$s + t(1 + |z|)^2 \leq 1 \,. \tag{35}$$

It is strong. For any $t$ value, it first restricts $|z|$ of interest to $\leq \frac{1}{\sqrt{t}} - 1$. Close to the upper limit means $s$ admissible has to be very small. So, the constraint may cut out quite a range, if not all, of the $s$ value of interest. We find that a solution exists in general, though some of the features of the solution locations are not somewhat peculiar and not easy to understand.

We scanned the effective coupling $G$ versus $s$, $|z|$, and $t$ plots to study the behavior of the intersecting point solutions and checked for consistency. The results are as follows: for somewhat large $t$, solution exists only at large enough $|z|$, for example the minimal $|z|$ value for solution at $t = 0.3$ is about 1. Such a solution certainly violates (35). Actually, solution satisfying the constraint shows up only for $t$ below about 0.265, which also guarantees the $G_+$ curve to be smooth, at least within the numerical window of interest. Moreover, the $G$ versus $|z|$ plots for any $t$ and $s$ essentially always give two solutions for (nonzero) $|z|$. The larger value $|z|$ solution may not even correspond to a larger coupling $G$, as shown in Figure 4. Furthermore, a $G$ value smaller than the $|z| = 0$ solution is typical. Another illustration of the same coupling value issue is given in Figure 5 in which we show $G$ versus $s$ plots with two intersecting points, particularly including one with $s = 0$. Such non-zero $|z|$ with $s = 0$ solutions are not available for $t$ less than about 0.17. For the latter case, the $G$ versus $s$ plots give a single intersecting point. In Figure 6, we show comparisons of the intersecting point solutions at the same $t$. We have again a solution with larger values of the parameter, $s$ and $|z|$, for the masses generated corresponding again to smaller coupling $G$. Recall the standard, obviously physical sensible, solution features of the NJL-type model which our $|z| = 0$ solutions shown above bears, is that nontrivial symmetry breaking mass solution exists for large enough coupling beyond a minimal critical value and increases with the coupling. The $|z| \neq 0$ 'solutions' behave, however, in ways difficult to understand. A more careful inspection of the various plots shows that the $G_-$ curve in particular has strange singularities. In fact, each intersecting point 'solution' corresponds to a pair of $s$ and $|z|$ values, with the $G_-$ curve either diverging at a smaller $s$ or at a smaller $|z|$ value. It sounds like in order to 'get' to that 'solution', one has to bring the coupling value all the way to positive or negative infinity and back. However, it should be noted that nonzero $\tilde{\eta}(= |z|\sqrt{t})$ increases the mass of one of the smaller mode but decreases that of the other one [*cf.* Equation (30)]. It is not so trivial to consider if larger $\tilde{\eta}$ or $|z|$ should really be considered to be giving a larger supersymmetry breaking effect. Another noteworthy feature is that among solutions of fixed $|z|$, a larger $t$ generally tends

to give smaller $s$ or $\tilde{m}^2$, and among solutions of fixed $s$, a larger $t$ generally tends to give larger $|z|$; larger $t$ always tends to increase coupling $G$ required for a solution. Recall that the $|m|$ or $t$ value also suppresses the mass generation in the $|z| = 0$ case, but $|m| = 0$ gives certainly no $|z| \neq 0$ solution.

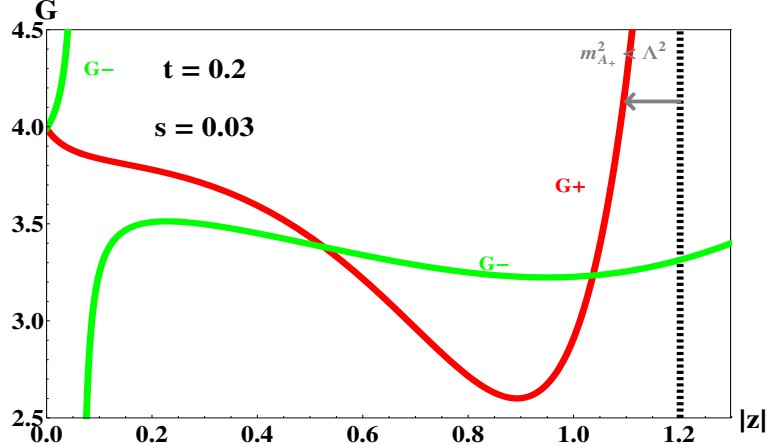

**Figure 4.** An illustrative of intersecting point solutions, with $G$ versus $|z|$.

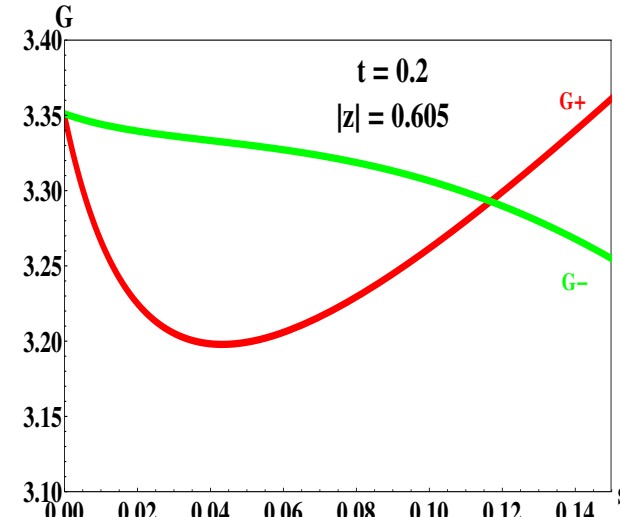

**Figure 5.** Illustrative intersecting point solution plots, with $G$ versus $s$.

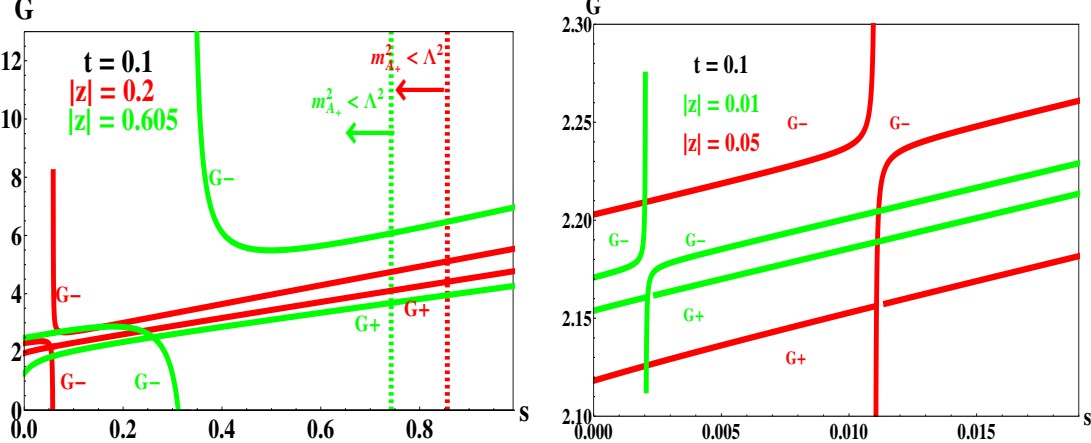

**Figure 6.** Illustrative intersecting point solution plots, with $G$ versus $s$.; two cases in each frame for comparison. The two colors each corresponds to the case of one set of fixed parameter values as shown. Intersecting points of $G^+$ and $G^-$ curves of the same color give the solution point for the value of $|z|$.

## 5. The Goldstino and Composite (Super) Field Dynamics

Some components of the superfield $U$, which are auxiliary as introduced, develop kinetic terms through wavefunction renormalizations in the effective theory below the cutoff $\Lambda$. We trace them here through checking of the relevant loop diagrams, based on the effective Lagrangian in terms of components of $\Phi_R$ and couplings all having the $\Phi$ wavefunction renormalization from the gap equation result incorporated (cf. equations in Appendix A). The analysis focuses on results at the supersymmetry breaking vacuum solutions, i.e. non-zero $\tilde{m}^2$ with zero or nonzero $\tilde{\eta}$. We only sketch the key results here, leaving some more details in Appendix A.

We start with the two-spinors $\chi$ and $\lambda$. The chirality conserving part of the self-energy diagrams gives rise to kinetic terms. We can see that all terms are nonzero in the presence of nonvanishing $\tilde{\eta}$, while the $\chi$-$\lambda$ kinetic mixing vanishes at $\tilde{\eta} = 0$. Full results are presented in Appendix A. To look at the mass values is complicated. One needs first to take a unitary transformation on the hermitian matrix and kinetic terms to diagonalize it. Denote the eigenvalues by $N_{f_1}$ and $N_{f_2}$, and the diagonalizing matrix by $T$. The canonically normalized fermionic modes are given by

$$
\begin{pmatrix} f_1 \\ f_2 \end{pmatrix} = \begin{pmatrix} \frac{1}{\sqrt{N_{f_1}}} & 0 \\ 0 & \frac{1}{\sqrt{N_{f_2}}} \end{pmatrix} T \begin{pmatrix} \chi \\ \lambda \end{pmatrix} .
\tag{36}
$$

Only the mass matrix for the canonical modes can be diagonalized to give the mass eigenvalues. The mass matrix $\mathcal{M}_f$ for $f_1$ and $f_2$ is hence given by

$$
\mathcal{M}_f = \begin{pmatrix} \sqrt{N_{f_1}} & 0 \\ 0 & \sqrt{N_{f_2}} \end{pmatrix} T \left( \mathcal{M}_{\chi\lambda} \right) T^T \begin{pmatrix} \sqrt{N_{f_1}} & 0 \\ 0 & \sqrt{N_{f_2}} \end{pmatrix} ,
\tag{37}
$$

where $-\mathcal{M}_{\chi\lambda} = \begin{pmatrix} 0 & -\mu \\ -\mu & 0 \end{pmatrix} + \Omega$, the first part being the tree-level mass while the last is the matrix for chirality-flipping pieces of self-energy diagrams. We have

$$
\det \mathcal{M}_f = N_{f_1} N_{f_2} \det \mathcal{M}_{\chi\lambda} .
\tag{38}
$$

In the case that the matrix of kinetic terms has the full rank, a zero determinant of $\det \mathcal{M}_f$ or equivalently $\det \mathcal{M}_{\chi\lambda}$ shows the existence of a Goldstino, which is expected as the Nambu–Goldstone fermion corresponding to the supersymmetry breaking. We are mostly interested only in the kind of qualitative questions here, which saves us from having to deal with the diagonalization of the matrix of kinetic terms. For the chirality-flipping diagrams (see Figure 7), dropping the $p$-dependent parts, we have the mass terms

$$
\begin{aligned}
\Omega_{\chi\chi} &= \frac{g^2 \tilde{m}^4}{\tilde{\eta}} |m|^2 N_c I_{3F}(|m|^2, m_{A_-}^2, m_{A_+}^2) - \frac{N_c}{2\tilde{\eta}} \left( g^2 I_C + \tilde{m}^2 g^2 I_{N'} \right) , \\
\Omega_{\chi\lambda} &= -2\mu g^2 \tilde{m}^2 |m|^2 N_c I_{3F}(|m|^2, m_{A_-}^2, m_{A_+}^2) + \mu g^2 N_c I_{N'} , \\
\Omega_{\lambda\lambda} &= \mu^2 g^2 \tilde{\eta} |m|^2 N_c I_{3F}(|m|^2, m_{A_-}^2, m_{A_+}^2) ,
\end{aligned}
\tag{39}
$$

where $I_{3F}(|m|^2, m_{A_-}^2, m_{A_+}^2)$ is the integral of the product of three Feynman propagators with the mass-squares as specified, and we have expressed the results with the $I_C$ and $I_{N'}$ integrals of the gap equations (cf. Equations (31) and (29)). Applications of the gap equations kill the term with the then vanishing $\left( g^2 I_C + \tilde{m}^2 g^2 I_{N'} \right)$ factor and has the $\mu g^2 N_c I_{N'}$ term canceling the tree-level term in the mass matrix $\mathcal{M}_{\chi\lambda}$, the determinant of which is then exactly zero. Hence, we have established the existence of a Goldstino mode for the supersymmetry breaking solution with $\tilde{\eta} \neq 0$. For the $\tilde{\eta} = 0$ case, only the off-diagonal terms are nonzero, which is a result one can see even simply from the $U(1)_R$

symmetry considerations. The latter has been presented in [16], with the result that the tree-level Dirac mass is exactly canceled by $\Omega$-matrix upon application of the corresponding gap equation, giving $\mathcal{M}_{\chi\lambda}$ as the zero matrix. No matter $m = 0$ or not, the qualitative feature does not change. Again, we have the Goldstino. Supersymmetry is really a local/spacetime symmetry. The Goldstino would be eaten up by the gravitino, which would then be massive.

The spin one vector boson $v^\mu$ is an important characteristic of the model. The proper self-energy diagrams (see Figure 8) for the vector mode give the result

$$
\begin{aligned}
\Sigma_v =\ & \mu^2 g^2 N_c \bigg\{ -|m|^2 I_{2F}(|m|^2, |m|^2) + \frac{1}{8} \int_0^1 dx \bigg[ \\
& \Big[ x(m_{A_-}^2 - m_{A_+}^2) - m_{A_-}^2 \Big] I_{2F}\left( -x(m_{A_-}^2 - m_{A_+}^2) + m_{A_-}^2, -x(m_{A_-}^2 - m_{A_+}^2) + m_{A_-}^2 \right) \\
& + I_F\left( -x(m_{A_-}^2 - m_{A_+}^2) + m_{A_-}^2 \right) \\
& + \Big[ x(m_{A_+}^2 - m_{A_-}^2) - m_{A_+}^2 \Big] I_{2F}\left( -x(m_{A_+}^2 - m_{A_-}^2) + m_{A_+}^2, -x(m_{A_+}^2 - m_{A_-}^2) + m_{A_+}^2 \right) \\
& + I_F\left( -x(m_{A_+}^2 - m_{A_-}^2) + m_{A_+}^2 \right) \bigg] \bigg\} \\
& + p^2 \frac{\mu^2 g^2 N_c}{2} \bigg\{ -\frac{1}{3} I_{2F}(|m|^2, |m|^2) \\
& - \frac{1}{4} \int_0^1 dx\, x(1-x) \Big[ I_{2F}\left( -x(m_{A_-}^2 - m_{A_+}^2) + m_{A_-}^2, -x(m_{A_-}^2 - m_{A_+}^2) + m_{A_-}^2 \right) \\
& + I_{2F}\left( -x(m_{A_+}^2 - m_{A_-}^2) + m_{A_+}^2, -x(m_{A_+}^2 - m_{A_-}^2) + m_{A_+}^2 \right) \Big] \bigg\} + \ldots \\
\overset{\bar{\eta}=0}{\longrightarrow}\ & \mu^2 g^2 N_c \bigg\{ -|m|^2 I_{2F}(|m|^2, |m|^2) + \frac{1}{4} \int_0^1 dx \Big[ -m_A^2 I_{2F}(m_A^2, m_A^2) + I_F(m_A^2) \Big] \bigg\} \\
& - p^2 \frac{\mu^2 g^2 N_c}{6} \bigg\{ I_{2F}(|m|^2, |m|^2) + \frac{1}{4} I_{2F}\left( m_A^2, m_A^2 \right) \bigg\},
\end{aligned}
\tag{40}
$$

with $I_{nF}$ denoting the integrals with product of $n$ Feynman propagators. There is also a tree-level mass-squared term of $-\frac{\mu^2}{2} v^\nu v_\nu$ to be added. It sure indicates that we have properly behaving kinetic and mass terms (note our metric convention).

The other scalar modes also acquire kinetic and mass terms accordingly. Mode mixings, however, make the result a lot less transparent. Details are given in Appendix A.

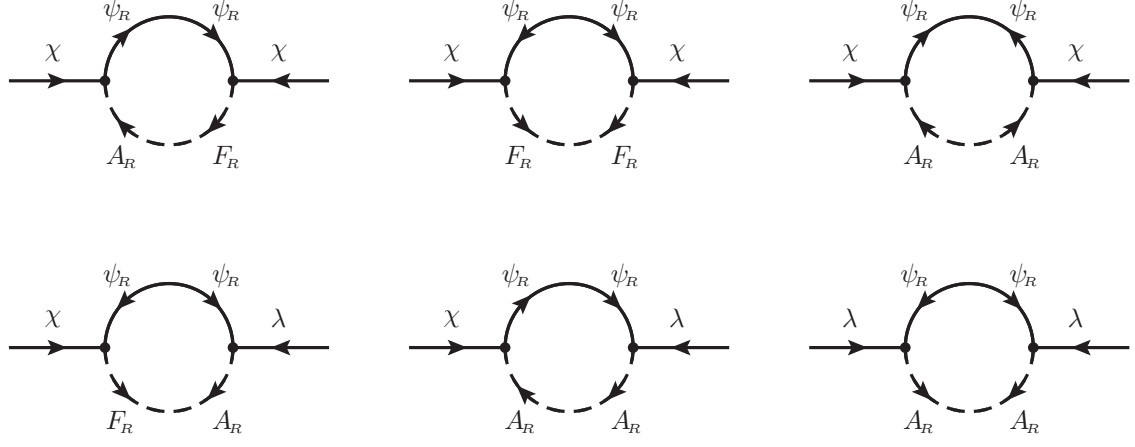

**Figure 7.** Diagrams for fermion masses.

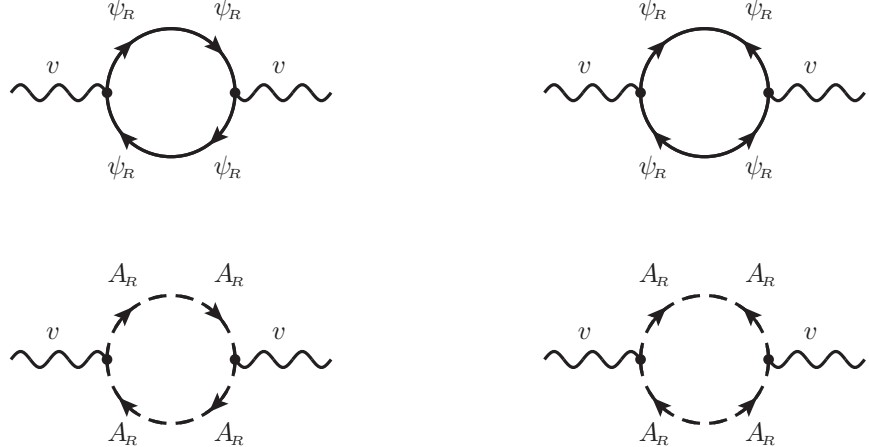

**Figure 8.** Proper self-energy diagrams for the spin one composite $v^\mu$.

## 6. Conclusions

We have studied the supersymmetric NJL model with a composite real superfield proposed in Reference [16] in a more general and complete scheme, by admitting a supersymmetric mass term for the chiral superfield while considering the most general form of the VEV for the composite superfield, hence allowing the soft supersymmetry breaking components of $\theta^2$ as well as $\theta^2\bar\theta^2$. Note that admitting the supersymmetric mass term actually reduces the symmetry of the model. The necessary conditions for the soft mass generation, and the basic features of the nontrivial solutions were discussed. We also calculated the contributions of all possible one-loop diagrams of component fields. The most general expressions for the superfield propagator were given in Appendix B. The analysis is much more technically involved compared to the previous limiting case though the basic approach is the same. However, the full analysis has quite interesting new features, as well as further nontrivial solutions.

The full analysis of all nontrivial vacuum solutions is difficult, and we have mostly to rely on numerical scanning of the multi-dimensional space for the parameters. There are some important general features to note. Firstly, no $\tilde\eta \neq 0$ solutions are possible for $m = 0$. The critical coupling bringing in plausible nontrivial solutions increases substantially with increasing $|m|$ in general. When the latter value gets beyond a certain limit, solutions with nontrivial soft mass solutions no longer exist. For nonzero $m$ within the limit, we find $\tilde\eta \neq 0$ solutions in general, from the numerical scanning. There are indications for an $\tilde\eta \neq 0$ solution to be preferred over the $\tilde\eta = 0$ in the sense that assuming fixed $m$ and $\tilde m^2$ values, the coupling value for an $\tilde\eta \neq 0$ solution seems to be always lower than the $\tilde\eta = 0$ one. For admissible solutions at a fixed coupling, however, the analysis cannot tell which solution would be more preferable.

We also calculated the contributions to the two-point functions for the component fields from all possible one-loop diagrams. All the scalar components become dynamical. The fermion mass matrix, while getting more complicated, clearly gives a zero eigenvalue for a Goldstino state. There is no indication of any problem. Some details and discussions are given in Appendix A. The results are useful for further analysis of the model and plausible applications. Meanwhile, it is interesting to note that applications of supersymmetric NJL models to condensed matter physics have been studied [23]. Studies of our model in the direction may also be considered.

Finally, we emphasize that with the modern effective (field) theory perspective, it is the most natural thing to consider any theory as an effective description of nature only within a limited domain/scale. Physics is arguably only about effective theories, as any theory can only be verified experimentally up to a finite scale, and there may always be a cut-off beyond that. Having a cutoff scale with the so-called nonrenormalizable higher dimensional operators is hence in no sense an undesirable feature. Model content not admitting any other parameter with mass dimension in the Lagrangian would be very natural. Here, we have illustrated a model that gives rise to nonzero values

for all admissible soft supersymmetry breaking masses for the single superfield, including $U(1)_R$ part, dynamically. Models of the kind may have a chance to be the source for the soft supersymmetry breaking in a supersymmetric standard model.

**Author Contributions:** Conceptualization, O.C.W.K.; methodology, O.C.W.K.; software, Y.-M.D.; formal analysis, Y.C., Y.-M.D., G.F. and O.C.W.K.; writing—original draft preparation, Y.C., G.F. and O.C.W.K.; writing—review and editing, Y.C., G.F., and O.C.W.K. All authors have read and agreed to the published version of the manuscript.

**Funding:** O.C.W.K. is partially supported by research grant 109-2112-M-008-016 from the MOST of Taiwan.

**Conflicts of Interest:** The authors declare no conflict of interest.

## Appendix A. Some More Technical Details of the Model Calculations

Starting with the effective Lagrangian of Equation (18) with $c$, $d$, and $n$ denoting VEVs of the (original) scalars $C$, $D$, and $N$, we have, in terms of the renormalized components $A_R = \sqrt{Z}A$, $\psi_R = \sqrt{Z}\psi$, and $F_R = \sqrt{Z}F$ of $\Phi_R = \sqrt{Z}\Phi$ with the common (supersymmetric) wavefunction renormalization factor $Z = 1 + g_o c$, the quadratic part of the Lagrangian is given by

$$
\begin{aligned}
\mathcal{L}_{eff}^{(2)} &= A_R^* \square A_R + i(\partial_\mu \bar{\psi}_R)\bar{\sigma}^\mu \psi_R + F_R^* F_R + \frac{m}{2}\left(2A_R F_R - \psi_R \psi_R\right) + \frac{m^*}{2}\left(2A_R^* F_R^* - \bar{\psi}_R \bar{\psi}_R\right) \\
&\quad + \mu CD - \mu\chi\lambda - \mu\bar{\chi}\bar{\lambda} + N\bar{N} - \frac{\mu^2}{2}v^\nu v_\nu + \mu g d A_R^* A_R + g n A_R F_R^* + g n^* A_R^* F_R \,,
\end{aligned}
\tag{A1}
$$

in which we have the renormalized mass and coupling $m = \frac{m_0}{Z}$ and $g = \frac{g_0}{Z}$. Here, the scalars $C$, $N$, and $D$ are the physical ones with VEVs already pulled out, though we do not distinguish them from the original ones with VEVs explicitly in notation. One can easily obtain the following propagator expressions:

$$
\begin{aligned}
\langle T(A_R A_R^*)\rangle &= \frac{-i(p^2 + |m|^2 + g^2|n|^2 - \mu g d)}{(p^2 + |m|^2 + g^2|n|^2 - \mu g d)^2 - 4g^2|n|^2|m|^2}\,, \\
\langle T(A_R A_R)\rangle &= \frac{2ign^* m^*}{(p^2 + |m|^2 + g^2|n|^2 - \mu g d)^2 - 4g^2|n|^2|m|^2}\,, \\
\langle T(F_R F_R^*)\rangle &= \frac{i(p^2 - \mu g d)(p^2 + |m|^2 + g^2|n|^2 - \mu g d)}{(p^2 + |m|^2 + g^2|n|^2 - \mu g d)^2 - 4g^2|n|^2|m|^2}\,, \\
\langle T(F_R F_R)\rangle &= \frac{-2ignm^*(p^2 - \mu g d)}{(p^2 + |m|^2 + g^2|n|^2 - \mu g d)^2 - 4g^2|n|^2|m|^2}\,, \\
\langle T(A_R F_R)\rangle &= \frac{im^*(p^2 + |m|^2 - g^2|n|^2 - \mu g d)}{(p^2 + |m|^2 + g^2|n|^2 - \mu g d)^2 - 4g^2|n|^2|m|^2}\,, \\
\langle T(A_R F_R^*)\rangle &= \frac{ign^*(p^2 - |m|^2 + g^2|n|^2 - \mu g d)}{(p^2 + |m|^2 + g^2|n|^2 - \mu g d)^2 - 4g^2|n|^2|m|^2}\,, \\
\langle T(\psi_{R_\alpha} \bar{\psi}_{R_{\dot{\beta}}})\rangle &= \frac{-ip_\mu \sigma^\mu_{\alpha\dot{\beta}}}{p^2 + |m|^2}\,, \\
\langle T(\psi_{R_\alpha} \psi_R^\beta)\rangle &= \frac{-im^* \delta_\alpha^\beta}{p^2 + |m|^2}\,.
\end{aligned}
\tag{A2}
$$

Note that $-\mu g d$ and $-gn$ here correspond to the (renormalized) soft mass terms $\tilde{m}^2$ and $\tilde{\eta}$. The propagator expressions can be matched to that of the superfield $\Phi$ in Equation (11).

The remaining, interaction, terms in the effective Lagrangian read

$$
\begin{aligned}
\mathcal{L}_{eff}^{int} \;=\;& gC\left[A_R^*\Box A_R + i(\partial_\mu\bar{\psi}_R)\bar{\sigma}^\mu\psi_R + F_R^*F_R\right] - \mu g\psi_R\lambda A_R^* - \mu g\bar{\psi}_R\bar{\lambda}A_R + \mu g D A_R^* A_R \\
& -i\frac{g}{2}\bar{\psi}_R\bar{\sigma}^\mu\chi\partial_\mu A_R + i\frac{g}{2}(\partial_\mu\bar{\psi}_R)\bar{\sigma}^\mu\chi A_R - g\chi\psi_R F_R^* + gN A_R F_R^* \\
& +i\frac{g}{2}\bar{\chi}\bar{\sigma}^\mu\psi_R\partial_\mu A_R^* - i\frac{g}{2}A_R^*\bar{\chi}\bar{\sigma}^\mu\partial_\mu\psi_R - g\bar{\chi}\bar{\psi}_R F_R + gN^* A_R^* F_R \\
& -\frac{\mu g}{\sqrt{2}}\eta^{\mu\nu}v_\mu i A_R^*\partial_\nu A_R + \frac{\mu g}{\sqrt{2}}\eta^{\mu\nu}v_\mu i(\partial_\nu A_R^*)A_R - \frac{\mu g}{\sqrt{2}}\eta^{\mu\nu}v_\mu\bar{\psi}_R\bar{\sigma}_\nu\psi_R\,.
\end{aligned}
\tag{A3}
$$

Note that the above gives essentially all parts of the Lagrangian, apart from a constant. The linear terms are canceled at the physical vacuum with consistent *c*, *n*, *d* solutions discussed in the main text.

In the following, we present some details of the 'quark-loop' contribution to the two-point functions for the various components of the composite superfield *U* at the supersymmetry breaking vacuum solutions, as discussed in Section 5. Though we argue in the text that $\tilde{\eta}\neq 0$ solution looks suspicious and is hard to understand, we present fully generic results for completion. The results may offer more insight into the problem.

The two-point functions for fermion kinetic terms are given by the diagrams in Figure A1, with the $ip\cdot\bar{\sigma}\,\Xi$ [24] results given as

$$
\begin{aligned}
\Xi_{\chi\chi} \;=\;& -\frac{g^2 N_c}{4}\Big[|m|\,(|m|-|\tilde{\eta}|)\,I_{2F}(|m|^2,m_{A_-}^2) + |m|\,(|m|+|\tilde{\eta}|)\,I_{2F}(|m|^2,m_{A_+}^2) \\
& +2I_F(m_{A_-}^2) - 2\left(2\tilde{m}^2+3|m|^2-2|\tilde{\eta}||m|\right)I_{2F}(m_{A_-}^2,m_{A_-}^2) \\
& +2I_F(m_{A_+}^2) - 2\left(2\tilde{m}^2+3|m|^2+2|\tilde{\eta}||m|\right)I_{2F}(m_{A_+}^2,m_{A_+}^2) \\
& -2|m|^2\left(m_{A_-}^2-2\tilde{m}^2-3|m|^2+2|\tilde{\eta}||m|\right)I_{3F}(|m|^2,m_{A_-}^2,m_{A_-}^2) \\
& -2|m|^2\left(m_{A_+}^2-2\tilde{m}^2-3|m|^2-2|\tilde{\eta}||m|\right)I_{3F}(|m|^2,m_{A_+}^2,m_{A_+}^2)\Big] + \cdots\,, \\
\xrightarrow{\tilde{\eta}=0}\;& -\frac{g^2 N_c}{2}\Big[2I_F(m_A^2) + |m|^2 I_{2F}(|m|^2,m_A^2) - 2\left(2\tilde{m}^2+3|m|^2\right)I_{2F}(m_A^2,m_A^2) \\
& +2|m|^2\left(\tilde{m}^2+2|m|^2\right)I_{3F}(|m|^2,m_A^2,m_A^2)\Big]\,,
\end{aligned}
\tag{A4}
$$

$$
\begin{aligned}
\Xi_{\chi\lambda} \;=\;& -\mu g^2\tilde{\eta}^* N_c\Bigg[\left(1-\frac{|m|}{4|\tilde{\eta}|}\right)I_{2F}(|m|^2,m_{A_-}^2) + \left(1+\frac{|m|}{4|\tilde{\eta}|}\right)I_{2F}(|m|^2,m_{A_+}^2) \\
& -m_{A_-}^2 I_{3F}(|m|^2,m_{A_-}^2,m_{A_-}^2) - m_{A_+}^2 I_{3F}(|m|^2,m_{A_+}^2,m_{A_+}^2)\Bigg] + \cdots\,, \\
\xrightarrow{\tilde{\eta}=0}\;& 0\,.
\end{aligned}
\tag{A5}
$$

$$
\begin{aligned}
\Xi_{\lambda\lambda} \;=\;& -\mu^2 g^2 N_c\Big[I_{2F}(m_{A_-}^2,m_{A_-}^2) + I_{2F}(m_{A_+}^2,m_{A_+}^2) \\
& -|m|^2 I_{3F}(|m|^2,m_{A_-}^2,m_{A_-}^2) - |m|^2 I_{3F}(|m|^2,m_{A_+}^2,m_{A_+}^2)\Big] + \cdots\,, \\
\xrightarrow{\tilde{\eta}=0}\;& -2\mu^2 g^2 N_c\left[I_{2F}(m_A^2,m_A^2) - |m|^2 I_{3F}(|m|^2,m_A^2,m_A^2)\right]\,,
\end{aligned}
\tag{A6}
$$

where $I_{nF}$ denote integrals each of a product of *n* Feynman propagators with the mass-square parameters as given. We have given, besides the general result also, the simplified expression at the $\tilde{\eta}=0$ limit. Recall

$$
m_{A_\mp}^2 = \tilde{m}^2 + (|m|\mp|\tilde{\eta}|)^2
$$

and at the limit $\tilde{\eta} = 0$, we have used $m_A^2 \equiv \tilde{m}^2 + |m|^2 = m_{A_-}^2 = m_{A_+}^2$. The vanishing kinetic mixing between $\chi$ and $\lambda$ can also be easily seen from $U(1)_R$ symmetry considerations.

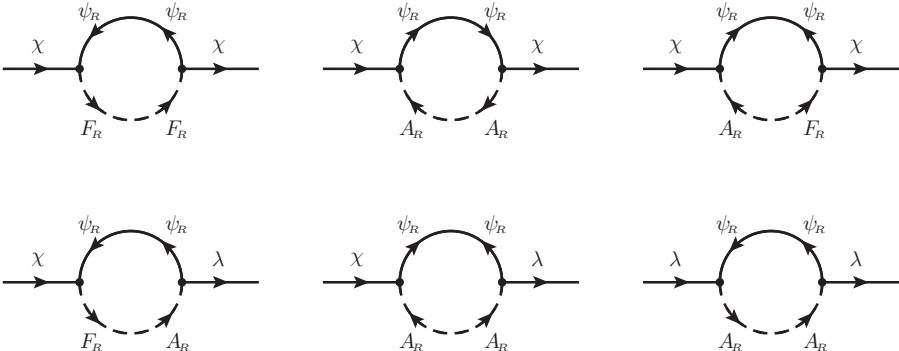

**Figure A1.** Diagrams for the generation of kinetic terms for the fermionic modes.

For the scalars, results for the various proper self-energy diagrams are very tedious. Apart from $I_{nF}$, we further introduce

$$I_{34}(m_a^2, m_b^2) \equiv 3I_{3F}(m_a^2, m_b^2, m_b^2) - 4m_b^2 I_{4F}(m_a^2, m_b^2, m_b^2, m_b^2) , \tag{A7}$$

where to present the results. Again, we give the general result and the $\tilde{\eta} = 0$ limit, with the diagrams being given in the Figures A2–A8. Note that they are one-loop contributions, which have to be summed up with the tree-level terms in the Lagrangian. We have:

$$
\begin{aligned}
\Sigma_{DD} = \;& p^2 \frac{\mu^2 g^2 N_c}{4} \left[ I_{34}(m_{A_-}^2, m_{A_-}^2) + I_{34}(m_{A_+}^2, m_{A_+}^2) \right] \\
& + \frac{\mu^2 g^2 N_c}{4} \left[ I_{2F}(m_{A_-}^2, m_{A_-}^2) + I_{2F}(m_{A_+}^2, m_{A_+}^2) \right] + \cdots , \\
& \xrightarrow{\tilde{\eta}=0} p^2 \frac{\mu^2 g^2 N_c}{2} I_{34}(m_A^2, m_A^2) + \frac{\mu^2 g^2 N_c}{2} I_{2F}(m_A^2, m_A^2) ,
\end{aligned}
\tag{A8}
$$

$$
\begin{aligned}
\Sigma_{CD} = \;& p^2 \frac{\mu g^2 N_c}{2} \left\{ \left[ m_{A_-}^2 + (|m| - |\tilde{\eta}|)^2 \right] I_{34}(m_{A_-}^2, m_{A_-}^2) + \left[ m_{A_+}^2 + (|m| + |\tilde{\eta}|)^2 \right] I_{34}(m_{A_+}^2, m_{A_+}^2) \right\} \\
& + \frac{\mu g^2 N_c}{2} \left\{ \left[ m_{A_-}^2 + (|m| - |\tilde{\eta}|)^2 \right] I_{2F}(m_{A_-}^2, m_{A_-}^2) + \left[ m_{A_+}^2 + (|m| + |\tilde{\eta}|)^2 \right] I_{2F}(m_{A_+}^2, m_{A_+}^2) \right. \\
& \left. - I_F(m_{A_-}^2) - I_F(m_{A_+}^2) \right\} + \cdots , \\
& \xrightarrow{\tilde{\eta}=0} p^2 \, \mu g^2 N_c \left( m_A^2 + |m|^2 \right) I_{34}(m_A^2, m_A^2) \\
& + \mu g^2 N_c \left[ \left( m_A^2 + |m|^2 \right) I_{2F}(m_A^2, m_A^2) - I_F(m_A^2) \right] ,
\end{aligned}
\tag{A9}
$$

$$
\begin{aligned}
\Sigma_{NN^*} = \;& p^2 \frac{g^2 N_c}{2} \left\{ (|m| - |\tilde{\eta}|)^2 I_{34}(m_{A_-}^2, m_{A_-}^2) + (|m| + |\tilde{\eta}|)^2 I_{34}(m_{A_+}^2, m_{A_+}^2) \right. \\
& \left. + |m| \, (|m| + |\tilde{\eta}|) \, I_{34}(m_{A_-}^2, m_{A_+}^2) + |m| \, (|m| - |\tilde{\eta}|) \, I_{34}(m_{A_+}^2, m_{A_-}^2) \right\} \\
& + \frac{g^2 N_c}{2} \left\{ (|m| - |\tilde{\eta}|)^2 I_{2F}(m_{A_-}^2, m_{A_-}^2) + (|m| + |\tilde{\eta}|)^2 I_{2F}(m_{A_+}^2, m_{A_+}^2) \right. \\
& \left. + 2|m|^2 I_{2F}(m_{A_-}^2, m_{A_+}^2) - I_F(m_{A_-}^2) - I_F(m_{A_+}^2) \right\} + \cdots , \\
& \xrightarrow{\tilde{\eta}=0} p^2 \, 2g^2 |m|^2 N_c I_{34}(m_A^2, m_A^2) + g^2 N_c \left[ 2|m|^2 I_{2F}(m_A^2, m_A^2) - I_F(m_A^2) \right] ,
\end{aligned}
\tag{A10}
$$

$$
\begin{aligned}
\Sigma_{CC} \;=\;& p^2 \frac{g^2 N_c}{8} \Big\{ 16|m|^2 \Big[ 3I_{2F}(|m|^2,|m|^2) - 4|m|^2 I_{3F}(|m|^2,|m|^2,|m|^2) \Big] - 16|m|^4 I_{34}(|m|^2,|m|^2) \\
& -6\left( m_{A_-}^2 + (|m|-|\tilde{\eta}|)^2 \right) I_{2F}(m_{A_-}^2, m_{A_-}^2) \\
& +8\left( m_{A_-}^2 + (|m|-|\tilde{\eta}|)^2 \right) m_{A_-}^2\, I_{3F}(m_{A_-}^2, m_{A_-}^2, m_{A_-}^2) \\
& -6\left( m_{A_+}^2 + (|m|+|\tilde{\eta}|)^2 \right) I_{2F}(m_{A_+}^2, m_{A_+}^2) \\
& +8\left( m_{A_+}^2 + (|m|+|\tilde{\eta}|)^2 \right) m_{A_+}^2\, I_{3F}(m_{A_+}^2, m_{A_+}^2, m_{A_+}^2) \\
& +2\left( m_{A_-}^2 + (|m|-|\tilde{\eta}|)^2 \right)^2 I_{34}(m_{A_-}^2, m_{A_-}^2) + m_{A_+}^2 \left( m_{A_-}^2 - m_{A_+}^2 \right) I_{34}(m_{A_-}^2, m_{A_+}^2) \\
& + m_{A_-}^2 \left( m_{A_+}^2 - m_{A_-}^2 \right) I_{34}(m_{A_+}^2, m_{A_-}^2) + 2\left( m_{A_+}^2 + (|m|+|\tilde{\eta}|)^2 \right)^2 I_{34}(m_{A_+}^2, m_{A_+}^2) \Big\} \\
& +\frac{g^2 N_c}{8} \Big\{ 24|m|^2 I_F(|m|^2) - 16|m|^4 I_{2F}(|m|^2,|m|^2) + \left( m_{A_+}^2 - 5m_{A_-}^2 - 8(|m|-|\tilde{\eta}|)^2 \right) I_F(m_{A_-}^2) \\
& - \left( 5m_{A_+}^2 - m_{A_-}^2 + 8(|m|+|\tilde{\eta}|)^2 \right) I_F(m_{A_+}^2) \\
& +2\left( m_{A_-}^2 + (|m|-|\tilde{\eta}|)^2 \right)^2 I_{2F}(m_{A_-}^2, m_{A_-}^2) \\
& + \left( 2m_{A_-}^2 m_{A_+}^2 - m_{A_+}^2 m_{A_+}^2 - m_{A_-}^2 m_{A_-}^2 \right) I_{2F}(m_{A_-}^2, m_{A_+}^2) \\
& +2\left( m_{A_+}^2 + (|m|+|\tilde{\eta}|)^2 \right)^2 I_{2F}(m_{A_+}^2, m_{A_+}^2) \Big\} + \cdots, \\[4pt]
\xrightarrow{\tilde{\eta}=0}\;& p^2 \frac{g^2 N_c}{2} \Big\{ 4|m|^2 \Big[ 3I_{2F}(|m|^2,|m|^2) - 4|m|^2 I_{3F}(|m|^2,|m|^2,|m|^2) \Big] - 4|m|^4 I_{34}(|m|^2,|m|^2) \\
& -3\left( m_A^2 + |m|^2 \right) I_{2F}(m_A^2, m_A^2) + 4\left( m_A^2 + |m|^2 \right) m_A^2 I_{3F}(m_A^2, m_A^2, m_A^2) \\
& + \left( m_A^2 + |m|^2 \right)^2 I_{34}(m_A^2, m_A^2) \Big\} \\
& +\frac{g^2 N_c}{2} \Big\{ 6|m|^2 I_F(|m|^2) - 4|m|^4 I_{2F}(|m|^2,|m|^2) - 2\left( m_A^2 + 2|m|^2 \right) I_F(m_A^2) \\
& + \left( m_A^2 + |m|^2 \right)^2 I_{2F}(m_A^2, m_A^2) \Big\}.
\end{aligned}
\tag{A11}
$$

There are more mixing terms which vanish with $\tilde{\eta}$, as follows:

$$
\begin{aligned}
\Sigma_{NN} \;=\;& p^2\, \frac{g^2 N_c}{4} \frac{\tilde{\eta}^2}{|\tilde{\eta}|^2} \Big[ (|m|-|\tilde{\eta}|)^2 I_{34}(m_{A_-}^2, m_{A_-}^2) + (|m|+|\tilde{\eta}|)^2 I_{34}(m_{A_+}^2, m_{A_+}^2) \\
& - |m|\,(|m|-|\tilde{\eta}|)\, I_{34}(m_{A_-}^2, m_{A_+}^2) - |m|\,(|m|+|\tilde{\eta}|)\, I_{34}(m_{A_+}^2, m_{A_-}^2) \Big] \\
& +\frac{g^2 N_c}{4} \frac{\tilde{\eta}^2}{|\tilde{\eta}|^2} \Big[ (|m|-|\tilde{\eta}|)^2 I_{2F}(m_{A_-}^2, m_{A_-}^2) + (|m|+|\tilde{\eta}|)^2 I_{2F}(m_{A_+}^2, m_{A_+}^2) \\
& -2|m|^2 I_{2F}(m_{A_-}^2, m_{A_+}^2) \Big],
\end{aligned}
\tag{A12}
$$

**Figure A2.** Proper self-energy diagrams for the *DD* term.

$$
\begin{aligned}
\Sigma_{CN^*} \;=\; & p^2\, \frac{g^2 N_c}{2}\, \frac{\tilde{\eta}}{|\tilde{\eta}|} \Big\{ (|m|+|\tilde{\eta}|)\left[ m_{A_+}^2 + (|m|+|\tilde{\eta}|)^2 \right] I_{34}(m_{A_+}^2,m_{A_+}^2) - m_{A_+}^2 |m| I_{34}(m_{A_-}^2,m_{A_+}^2) \\
& + m_{A_-}^2 |m| I_{34}(m_{A_+}^2,m_{A_-}^2) - (|m|-|\tilde{\eta}|)\left[ m_{A_-}^2 + (|m|-|\tilde{\eta}|)^2 \right] I_{34}(m_{A_-}^2,m_{A_-}^2) \Big\} \\
& + \frac{g^2 N_c}{2}\, \frac{\tilde{\eta}}{|\tilde{\eta}|} \Big\{ (3|m|-2|\tilde{\eta}|)\, I_F(m_{A_-}^2) - (3|m|+2|\tilde{\eta}|)\, I_F(m_{A_+}^2) - 4|m|^2|\tilde{\eta}|\, I_{2F}(m_{A_-}^2,m_{A_+}^2) \\
& - (|m|-|\tilde{\eta}|)\left[ m_{A_-}^2 + (|m|-|\tilde{\eta}|)^2 \right] I_{2F}(m_{A_-}^2,m_{A_-}^2) \\
& + (|m|+|\tilde{\eta}|)\left[ m_{A_+}^2 + (|m|+|\tilde{\eta}|)^2 \right] I_{2F}(m_{A_+}^2,m_{A_+}^2) \Big\}\, ,
\end{aligned}
\tag{A13}
$$

$$
\begin{aligned}
\Sigma_{DN^*} \;=\; & p^2\, \frac{\mu g^2 N_c}{2}\, \frac{\tilde{\eta}}{|\tilde{\eta}|} \Big[ (|m|+|\tilde{\eta}|)\, I_{34}(m_{A_+}^2,m_{A_+}^2) - (|m|-|\tilde{\eta}|)\, I_{34}(m_{A_-}^2,m_{A_-}^2) \\
& - |m| I_{34}(m_{A_-}^2,m_{A_+}^2) + |m| I_{34}(m_{A_+}^2,m_{A_-}^2) \Big] \\
& + \frac{\mu g^2 N_c}{2}\, \frac{\tilde{\eta}}{|\tilde{\eta}|} \Big\{ (|m|+|\tilde{\eta}|)\, I_{2F}(m_{A_+}^2,m_{A_+}^2) - (|m|-|\tilde{\eta}|)\, I_{2F}(m_{A_-}^2,m_{A_-}^2) \Big\}\, ,
\end{aligned}
\tag{A14}
$$

with also the complex conjugates for the last three, i.e., $-\Sigma_{N^*N^*}$, $-\Sigma_{CN}$, and $-\Sigma_{DN}$.

It is somewhat surprising that all the scalar actually becomes dynamic, including $D$ and $N$. The latter are introduced as auxiliary components of mass dimension two. One should hence consider $\frac{D}{\mu}$ and $\frac{N}{\mu}$ instead. For the general case, the complex $\frac{N}{\mu}$ has to be expanded into the real components first. One has then to diagonalize the kinetic term matrix for all the real scalars to find the proper wavefunction renormalization factors for the canonical modes, and subsequently diagonalize the mass-square matrix, with the tree-level terms included, of the latter for the eigenvalues.

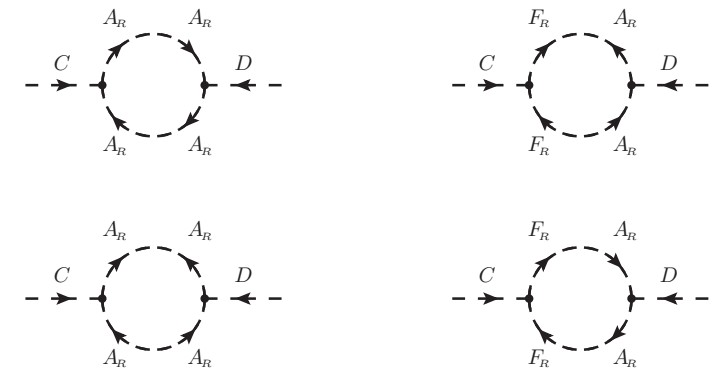

**Figure A3.** Proper self-energy diagrams for the $CD$ term.

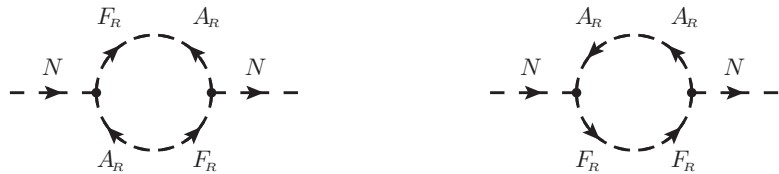

**Figure A4.** Proper self-energy diagrams for the $NN^*$ term.

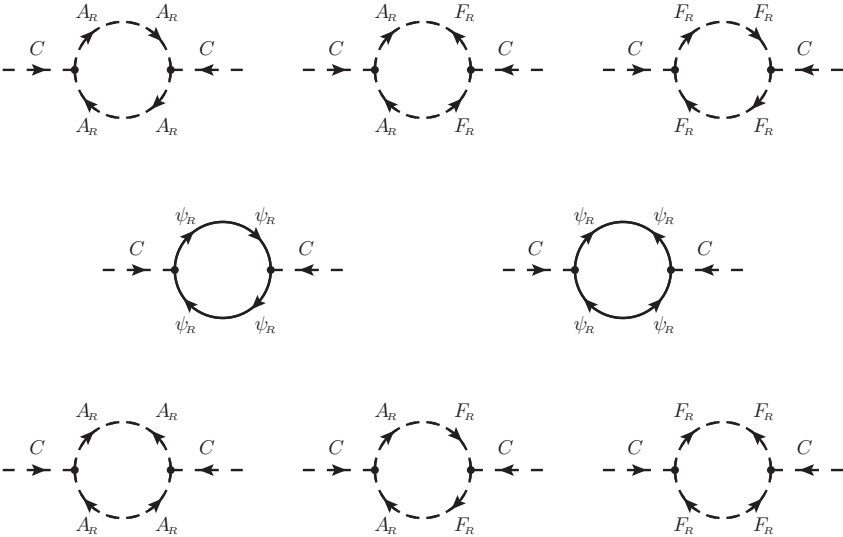

**Figure A5.** Proper self-energy diagrams for the $CC$ term.

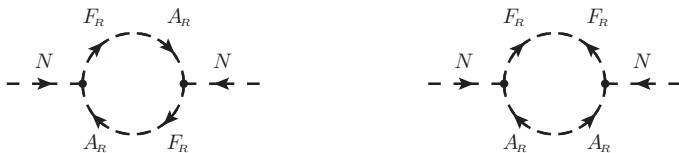

**Figure A6.** Proper self-energy diagrams for the $NN$ term.

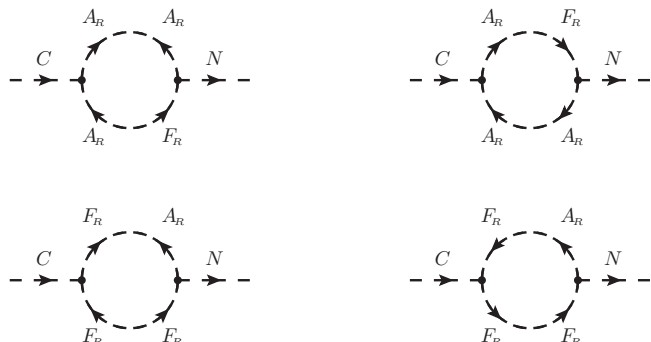

**Figure A7.** Proper self-energy diagrams for the $CN^*$ term.

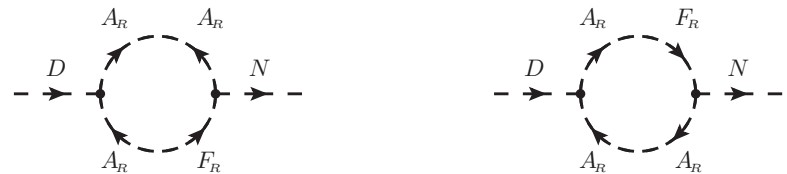

**Figure A8.** Proper self-energy diagrams for the $DN^*$ term.

## Appendix B. Propagator Expressions for the Most General Case

We give here the superfield propagator expressions for the most general case, i.e., all soft supersymmetry breaking parameters are included. The propagator(s) used in our model above is the case with the soft mass term $-\frac{1}{2}\eta\theta^2\Phi^2$ in the superpotential vanishing. The expressions have, apparently, not been explicitly given before, and may be useful in some future studies.

The free-field Lagrangian for a single chiral superfield $\Phi = A + \sqrt{2}\psi\theta + F\theta^2$ admitting all supersymmetric and (soft) supersymmetry breaking mass parameters can be written as

$$\mathcal{L}_o = \int d^4\theta \; \bar{\Phi}\Phi(1 - \tilde{\eta}\theta^2 - \tilde{\eta}^*\bar{\theta}^2 - \tilde{m}^2\theta^2\bar{\theta}^2) + \left[\int d^2\theta \; \frac{1}{2}(m - \eta\theta^2)\Phi^2\delta^2(\bar{\theta}) + h.c.\right]. \quad (A15)$$

Again, we allow a complex $m$. Soft supersymmetry breaking parameters $\tilde{\eta}$ and $\eta$ are also complex while the most familiar soft mass $\tilde{m}^2$ is real. The superfield propagators are given by

$$\langle T(\Phi(1)\Phi^\dagger(2))\rangle = \frac{-i}{p^2 + |m|^2}\delta_{12}^4 - \frac{i[\tilde{\eta}(Q - 2|m|^2) + m^*\eta]}{Q^2 - |\eta - 2m\tilde{\eta}|^2}\theta_1^{\,2}\delta_{12}^4 - \frac{i[\tilde{\eta}^*(Q - 2|m|^2) + m\eta^*]}{Q^2 - |\eta - 2m\tilde{\eta}|^2}\bar{\theta}_1^{\,2}\delta_{12}^4$$

$$+ i\frac{(-p^2|\tilde{\eta}|^2 + \tilde{m}^2|m|^2)Q + 4p^2|m|^2|\tilde{\eta}|^2 - (p^2 - |m|^2)(m^*\eta\tilde{\eta}^* + m\eta^*\tilde{\eta}) - |m|^2|\eta|^2}{(p^2 + |m|^2|)(Q^2 - |\eta - 2m\tilde{\eta}|^2)}\theta_1^2\bar{\theta}_1^{\,2}\delta_{12}^4$$

$$+ i\frac{(\tilde{m}^2 + |\tilde{\eta}|^2)Q - |\eta - 2m\tilde{\eta}|^2}{(p^2 + |m|^2)(Q^2 - |\eta - 2m\tilde{\eta}|^2)}\left[\frac{D_1^2\theta_1^2\bar{\theta}_1^{\,2}\overline{D}_1^{\,2}}{16}\right]\delta_{12}^4 \;, \quad (A16)$$

and

$$\langle T(\Phi(1)\Phi(2))\rangle = \frac{i\,m^*}{p^2(p^2 + |m|^2)}\frac{D_1^2}{4}\delta_{12}^4 - \frac{i(\eta^* - 2m^*\tilde{\eta}^*)}{Q^2 - |\eta - 2m\tilde{\eta}|^2}\frac{D_1^2\bar{\theta}_1^{\,2}}{4}\delta_{12}^4$$

$$+ i\,\frac{2m^*\tilde{\eta}(p^2 + \tilde{m}^2) + m^{*2}\eta + \eta^*\tilde{\eta}^2}{Q^2 - |\eta - 2m\tilde{\eta}|^2}\frac{D_1^2\theta_1^2}{4p^2}\delta_{12}^4$$

$$+ i\,\frac{m^*[(\tilde{m}^2 + |\tilde{\eta}|^2)Q - |\eta - 2m\tilde{\eta}|^2] - \tilde{\eta}(\eta^* - 2m^*\tilde{\eta}^*)(p^2 + |m|^2)}{(p^2 + |m|^2)(Q^2 - |\eta - 2m\tilde{\eta}|^2)}\left[\frac{D_1^2\theta_1^2\bar{\theta}_1^{\,2}}{4} + \frac{\bar{\theta}_1^{\,2}\theta_1^2 D_1^2}{4}\right]\delta_{12}^4. \quad (A17)$$

where $Q = p^2 + |m|^2 + \tilde{m}^2 + |\tilde{\eta}|^2$. The corresponding component field propagators are given by

$$\langle T(A\,A^*)\rangle = \frac{-i(p^2 + |m|^2 + \tilde{m}^2 + |\tilde{\eta}|^2)}{(p^2 + |m|^2 + \tilde{m}^2 + |\tilde{\eta}|^2)^2 - |\eta - 2m\tilde{\eta}|^2} \;,$$

$$\langle T(A\,A)\rangle = \frac{i(\eta^* - 2m^*\tilde{\eta}^*)}{(p^2 + |m|^2 + \tilde{m}^2 + |\tilde{\eta}|^2)^2 - |\eta - 2m\tilde{\eta}|^2} \;,$$

$$\langle T(F\,F^*)\rangle = \frac{i[(p^2 + \tilde{m}^2)(p^2 + |m|^2 + \tilde{m}^2 + |\tilde{\eta}|^2) - |\eta - m\tilde{\eta}|^2 + |m\tilde{\eta}|^2]}{(p^2 + |m|^2 + \tilde{m}^2 + |\tilde{\eta}|^2)^2 - |\eta - 2m\tilde{\eta}|^2} \;,$$

$$\langle T(F\,F)\rangle = \frac{i[2m^*\tilde{\eta}(p^2 + \tilde{m}^2) + m^{*2}\eta + \eta^*\tilde{\eta}^2]}{(p^2 + |m|^2 + \tilde{m}^2 + |\tilde{\eta}|^2)^2 - |\eta - 2m\tilde{\eta}|^2} \;,$$

$$\langle T(A\,F)\rangle = \frac{i[m^*(p^2 + |m|^2 + \tilde{m}^2 - |\tilde{\eta}|^2) + \eta^*\tilde{\eta}]}{(p^2 + |m|^2 + \tilde{m}^2 + |\tilde{\eta}|^2)^2 - |\eta - 2m\tilde{\eta}|^2} \;,$$

$$\langle T(AF^*)\rangle = \frac{-i[\tilde{\eta}^*(p^2 - |m|^2 + \tilde{m}^2 + |\tilde{\eta}|^2) + m\eta^*]}{(p^2 + |m|^2 + \tilde{m}^2 + |\tilde{\eta}|^2)^2 - |\eta - 2m\tilde{\eta}|^2} \;,$$

$$\langle T(\psi_{R_\alpha}\bar{\psi}_{R_{\dot{\beta}}})\rangle = \frac{-ip_\mu \sigma^\mu_{\alpha\dot{\beta}}}{p^2 + |m|^2} \;,$$

$$\langle T(\psi_{R_\alpha}\psi_R^\beta)\rangle = \frac{-im^*\delta_\alpha^\beta}{p^2 + |m|^2} \;. \quad (A18)$$

Note that the Lagrangian without all the masses has a $U(1)$ and a $U(1)_R$ symmetry to which $\Phi$ carries both charges (of 1). $U(1)_R$ charges for the components $A$ and $F$ are 1 and $-1$, with $\psi$ neutral. We can assign corresponding charges to the mass parameters and use them to trace and check the role of the parameters in the component field propagators and the corresponding terms of the superfield propagators.

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
