# Peer review of "Further Studies of the Supersymmetric NJL-Type Model for a Real Superfield Composite"

_symmetry, doi:10.3390/sym12111818_

Round 1

Reviewer 1 Report

This paper looks at additional studies of a supersymetric NJL model. Here the authors now add a mss term and phi squared term. The look at various ground state solutions and low energy states. The paper seems easy enough to read and fits in nicely with scope of the journal in terms of symmetry. The results are focused on high energy physics but perhaps the authors could say something about possible applications to condensed matter condensates or if a supersymmetric BES could even be created.

Author Response

We thank the reviewer for the comments and suggestions on the issue of its applications. In the literature there were indeed studies discussing the applications of supersymmetric NJL models to condensed matter physics. However, the area is beyond our scope of studies and expertise. At this point we don't think we can do much beyond citing the possibility, which we added in the modified manuscript.

Reviewer 2 Report

The authors study a supersymmetric NJL-type model for a real superfield composite. They generalize their earlier analysis by allowing a supersymmetric mass term for the chiral superfield, as well as possible components for the soft supersymmetry breaking part of the condensate. The paper discusses in great detail the supersymmetric standard model and symmetry breaking terms.

The language of the paper may be improved. There is a number of misprints that should be revised before publication, e.g. the authors do use a variable of integration in Eq.(1) and Eq.(14), but do not bother to mention it in Eqs. (4-6). I can suggest that to make a paper more readable to add some general definitions for the main terms used in the paper, like “Goldstino is the Nambu-Goldstone fermion emerging in the spontaneous breaking of supersymmetry”.

To summarize, I can recommend this paper for publication after minor revision.

Author Response

We thank the reviewer for the careful study and helpful comments. We have put the variable (d^4\theta) of integration back to equation (4)-(6), to make the whole manuscript consistent on the issue. Some more explanations to the main terms (e.g., Goldstino, chiral superfield, Grassmann number) are added to make it easier to read. Typos and misprints are also corrected as we can find.